# Identification of anticancer drugs for hepatocellular carcinoma through personalized genome-scale metabolic modeling

Rasmus Agren[1,†], Adil Mardinoglu[1,†], Anna Asplund[2], Caroline Kampf[2], Mathias Uhlen[3,4] & Jens Nielsen[1,3,*]

## Abstract

*Genome-scale metabolic models (GEMs) have proven useful as scaffolds for the integration of omics data for understanding the genotype–phenotype relationship in a mechanistic manner. Here, we evaluated the presence/absence of proteins encoded by 15,841 genes in 27 hepatocellular carcinoma (HCC) patients using immunohistochemistry. We used this information to reconstruct personalized GEMs for six HCC patients based on the proteomics data, HMR 2.0, and a task-driven model reconstruction algorithm (tINIT). The personalized GEMs were employed to identify anticancer drugs using the concept of antimetabolites; i.e., drugs that are structural analogs to metabolites. The toxicity of each antimetabolite was predicted by assessing the in silico functionality of 83 healthy cell type-specific GEMs, which were also reconstructed with the tINIT algorithm. We predicted 101 antimetabolites that could be effective in preventing tumor growth in all HCC patients, and 46 antimetabolites which were specific to individual patients. Twenty-two of the 101 predicted antimetabolites have already been used in different cancer treatment strategies, while the remaining antimetabolites represent new potential drugs. Finally, one of the identified targets was validated experimentally, and it was confirmed to attenuate growth of the HepG2 cell line.*

**Keywords** antimetabolites; genome-scale metabolic models; hepatocellular carcinoma; personalized medicine; proteome
**Subject Categories** Genome-Scale & Integrative Biology; Pharmacology & Drug Discovery
**Mol Syst Biol (2014) 10: 721**

## Introduction

Hepatocellular carcinoma (HCC) is the most common form of primary liver cancer and affects more than half a million people worldwide, with the highest incidences in Asia and Africa (Jemal *et al*, 2011). It is the third leading cause of cancer death, and the global burden of HCC continues to increase worldwide (Finn, 2010). Multiple etiologic factors are implicated in the development of HCC (Sanyal *et al*, 2010), and these factors have a direct impact on patient characteristics as well as the tumor progression. The treatment of an HCC patient is dependent on the size, stage, and grade of the tumor, and several treatment modalities are available, including liver transplantation and local ablative therapies (Padhya *et al*, 2013). There is an urgent need for the development of efficient drugs since sorafenib, an approved drug for HCC, is effective only in approximately 30% of the patients (Bruix & Sherman, 2011). The diagnosis of HCC still remains difficult and there are large gaps in our current understanding about the underlying molecular mechanisms involved in the pathogenesis of HCC (Sanyal *et al*, 2010). The elucidation of these diverse mechanisms for the identification of novel drug targets has therefore been a major focus in medicine, and further research efforts are still needed for an increased understanding and for developing efficient treatment strategies. However, this is quite a challenging task, since HCC involves a large number of interplays between different biological pathways (Ye *et al*, 2012). In addition to environmental factors, individual genetic variations should also be accounted for developing effective treatment strategies and for optimizing surveillance methods. This could be achieved through personalized medicine (Chen *et al*, 2012; Hood *et al*, 2012), a novel approach to healthcare that allows to tailor the treatment strategy based on the individual's genetic makeup (Hood & Friend, 2011).

The decreasing cost of omics profiling has made high-throughput technologies available for understanding the molecular basis of disease. Genome-scale metabolic models (GEMs) can aid in this by providing a scaffold for the integration of omics data

1  Department of Chemical and Biological Engineering, Chalmers University of Technology, Gothenburg, Sweden
2  Department of Immunology, Genetics and Pathology Science for Life Laboratory, Uppsala University, Uppsala, Sweden
3  Science for Life Laboratory KTH – Royal Institute of Technology, Stockholm, Sweden
4  Department of Proteomics KTH – Royal Institute of Technology, Stockholm, Sweden
  *Corresponding author. Tel: +46 31 772 3804; Fax: +46 31 772 3801; E-mail: nielsenj@chalmers.se
  †These authors contributed equally to this work.

(Mardinoglu & Nielsen, 2012). To date, different generic human GEMs (Duarte *et al*, 2007; Mardinoglu *et al*, 2013a, 2014; Thiele *et al*, 2013) have been reconstructed. These models can be successfully employed to gain further biological and mechanistic understanding of metabolism-related diseases, discovering potential biomarkers and identifying novel drug targets (Mardinoglu *et al*, 2013b; Väremo *et al*, 2013). Furthermore, different GEMs for cancer have been reconstructed to characterize the genetic mechanisms of cancer and to reveal how cancer cells benefit from metabolic modifications (Folger *et al*, 2011; Frezza *et al*, 2011; Agren *et al*, 2012). In particular, Folger *et al* (2011) predicted 52 potential cytostatic anticancer drug targets and synthetic lethal gene targets through gene knockdowns by employing a generic cancer GEM.

Cancer cells proliferate rapidly and adapt their metabolism based on the availability of nutrients necessary for the synthesis of new building blocks, a feature which is the basis for many of the anticancer drugs currently in use (Lazar & Birnbaum, 2012). One of the most common types of anticancer drugs is antimetabolites which prevent the use of one or more endogenous metabolites by inhibiting the catalyzing enzymes, normally by being structurally similar to the metabolite(s) in question. Examples of such antimetabolites are antifolates and antipurines, which mimic folic acid and purines, respectively. This type of drugs has been used for decades and results in disrupted robustness of the cancer cell and reduction or suppression of growth (Garg *et al*, 2010; Hebar *et al*, 2013). The purpose of this study was to predict potential antimetabolites (or rather, their corresponding endogenous metabolites) for HCC by simulating their effect using genome-scale metabolic modeling.

In this study, we first evaluated the presence/absence of proteins encoded by 15,841 genes in tumors obtained from 27 HCC patients (Fig 1A) and identified extensive differences between six HCC patients based on the proteomics data. Secondly, we developed the tINIT (Task-driven Integrative Network Inference for Tissues) algorithm, which allows for automated reconstruction of functional GEMs based on protein evidence and a novel task-driven reconstruction approach (Fig 1B). To this end, we defined a set of core metabolic tasks that should occur in all cell types and enforced their functionality during the GEM reconstruction process. Thirdly, we applied tINIT to the Human Metabolic Reaction database (HMR) 2.0 (Mardinoglu *et al*, 2014) together with personalized proteomics data and reconstructed functional personalized GEMs for six HCC patients. Fourthly, we generated functional GEMs for 83 different healthy cell types based on the proteomic data in the Human Protein Atlas (HPA, www.proteinatlas.org). This approach represents a significant step forward in the reconstruction of cell type-specific models, given that tINIT not only generates connected and consistent metabolic networks, but also ensures functionality by integrating evidence-based metabolic functions which are established for a certain cell type. Furthermore, we identified potential antimetabolites that were predicted to inhibit or kill the growth of HCC tumors in all six patients (Fig 1C). Since the inhibition of a metabolite may lead to abnormalities in the metabolic functions of a healthy cell, the toxic effect of each antimetabolite was evaluated for a number of metabolic tasks in each of the 83 healthy cell type GEMs. The proposed antimetabolites are therefore likely to damage the tumor in all HCC patients, with the least possible side effects on the function of other healthy cell types (i.e., having high efficacy and low toxicity). Finally, we experimentally evaluated the effect of an

L-carnitine analog, one of the predicted antimetabolites for the inhibition of HCC tumor growth in all patients. By evaluating proliferation of HepG2 cell lines in the presence or absence of the analog, we could confirm our genome-scale modeling predictions. The presented method allowed for the identification of potential therapeutic targets for treatment of HCC by considering individual differences in protein expression patterns and may therefore aid in filling the existing gap between proteomics and drug discovery.

## Results

### Personalized proteome data for HCC patients

The presence/absence of proteins encoded by 15,841 genes (Supplementary Dataset S1) in HCC tumor obtained from 27 male and female HCC patients (Supplementary Dataset S2) was evaluated using 18,707 antibodies generated in the HPA project. Duplicate evaluations of every protein were performed, and the abundance of each protein was analyzed in three or more HCC patients. The proteomics data displayed notable differences between the 27 HCC patients during the determination of the global protein expression levels (Fig 2A).

Here, we focused on the proteomics differences of six HCC patients that had the largest coverage of protein expression levels (Fig 2B). The number of the evaluated proteins in these HCC patients varied between 9,312 and 14,561. The expression levels of 4,936 proteins were identified in all six HCC patients and healthy hepatocytes (Fig 2C and Supplementary Dataset S3). Figure 2C demonstrates the apparent differences between the proteomics data of the HCC tumors and with the healthy hepatocytes. Functional differences between these 4936 proteins were identified through the level 5 gene ontology biological process (GO BP) terms (Supplementary Dataset S4) using DAVID (Huang *et al*, 2009) and GO BP terms with adjusted $P$-values < 0.005 are presented in Supplementary Fig S1.

In order to reconstruct personalized GEM for the six HCC patients based on the proteomics data, the missing expression of protein in each patient was predicted based on the protein expressions in the other 26 HCC patients. Hereby, the absence/presence of 15,841 proteins could be evaluated in the six HCC patients (Supplementary Dataset S5) and these proteomics data for each HCC patient were used for the reconstruction of personalized GEMs. The functional differences of the proteomics data were also identified through GO BP terms (Supplementary Dataset S6) using DAVID (Huang *et al*, 2009) and GO BP terms with adjusted $P$-values < 0.0001 are presented in Supplementary Fig S2. It is observed that there are significant differences in GO BP terms between the hepatocytes and the HCC patients. In particular, GO BP terms including positive regulation of programmed cell death, positive regulation of apoptosis, mitosis, M phase of mitotic cell cycle, protein catabolic process, fatty acid (FA) catabolic process, FA oxidation and lipid oxidation showed notable changes in all HCC patients compared to healthy hepatocytes.

### Personalized GEMs for HCC patients

It is known that the synthesis, degradation, and redistribution of metabolites and minerals as well as metabolite consumption rates

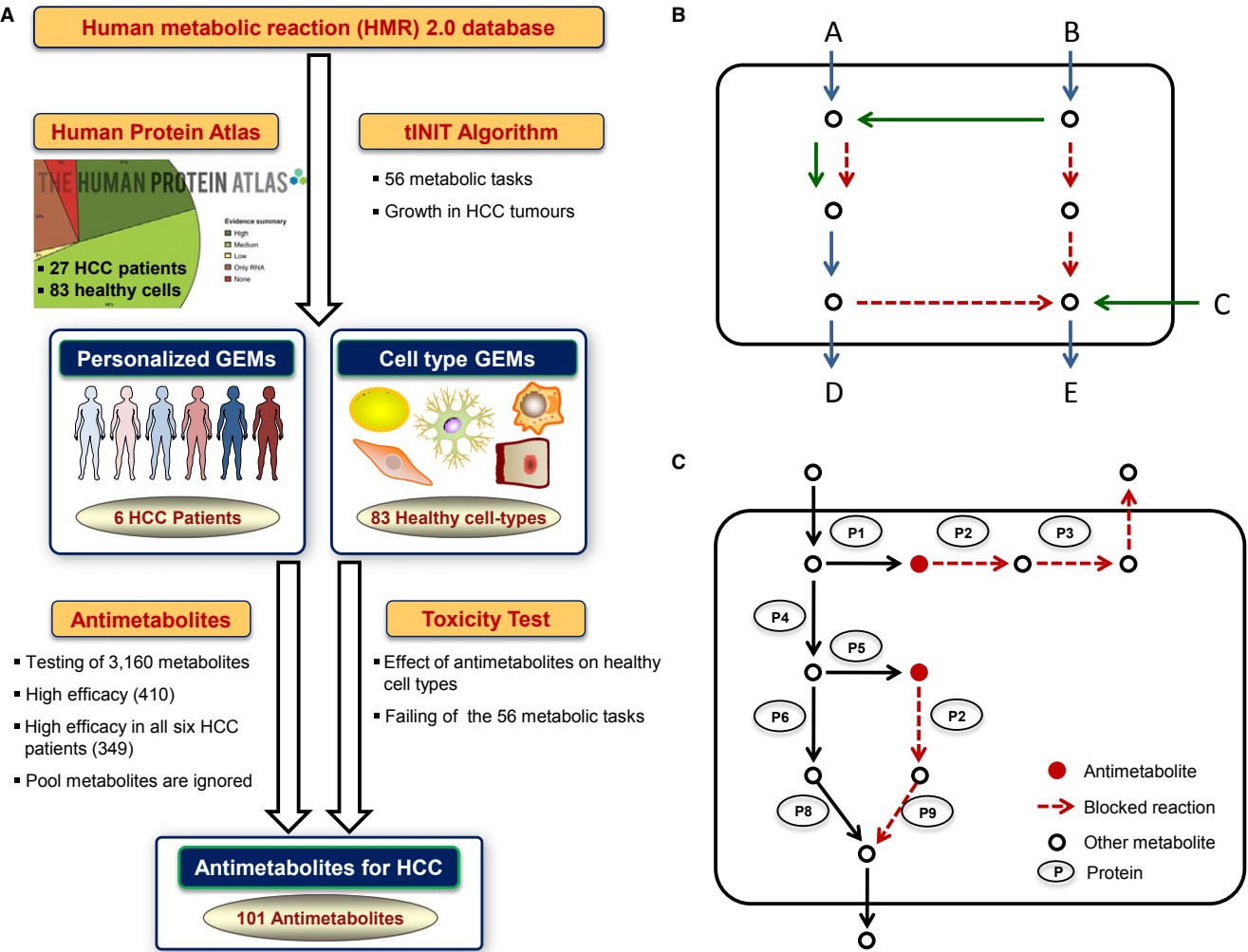

**Figure 1. General pipeline for the identification of antimetabolites.**

A   The presence/absence of 15,841 proteins in tumors obtained from 27 HCC patients was evaluated using immunohistochemistry. tINIT algorithm was developed and used for reconstruction of personalized GEMs for six HCC patients and GEMs for 83 healthy cell types based on proteomics data and HMR 2.0. A method identifying potential antimetabolites for the treatment of the HCC patients was developed, and the toxicity of each antimetabolite was predicted using GEMs for healthy cells types.

B   tINIT was used for reconstructing GEMs which are in agreement with omics data and which could perform a set of predefined tasks. In this illustration, the model should perform two simple tasks; production of "D" from "A" and "E" from "B." The resulting model should contain as many of the green reactions and as few of the red ones as possible. In the first step, all reactions were identified which, if removed from the network, cause any of the tasks to fail. These reactions were marked blue. In the second step, the INIT algorithm was used to find the network with the maximal number of green reactions compared to red, with the additional constraints that the model must be functional and that all blue reactions must be included. This would result in the dotted reactions being removed from the network. At this stage, the first task would be possible, but not the second one (since uptake of "C" makes the production of "E" possible without using any red reactions). In the final step, each task was tested and a gap-filling algorithm was used to reinsert the reactions which were required for all tasks to work. This would result in the inclusion of the lower-most red reaction.

C   The effect of antimetabolites can be predicted *in silico* by using metabolic network and potential use of antimetabolites is illustrated.

are different in cancer versus normal cell types (Lazar & Birnbaum, 2012). We have previously developed the INIT algorithm and used it to reconstruct active metabolic networks for 69 cell types and 16 cancers (Agren *et al*, 2012). We identified the common metabolic differences between cell types and cancers using the reconstructed model. These networks could be seen as snapshots of active metabolism in a given cell type, but they were not functional models that could be directly applied for simulations. Recently, we constructed HMR 2.0 (Mardinoglu *et al*, 2014), which contains 8,181 reactions,

6,007 metabolites in eight different compartments, and 3,765 genes associated with the reactions (Table 1).

Here, we reconstructed personalized GEMs for six HCC patients based on personalized proteomics data and HMR 2.0 by employing the new concept of task-driven model reconstruction (Fig 1B). This was done by first defining a list of 56 metabolic tasks which must occur in all cell types and that the resulting model should be able to perform (Supplementary Dataset S7). Literature evidence for the occurrence of these tasks in all human cells is provided in

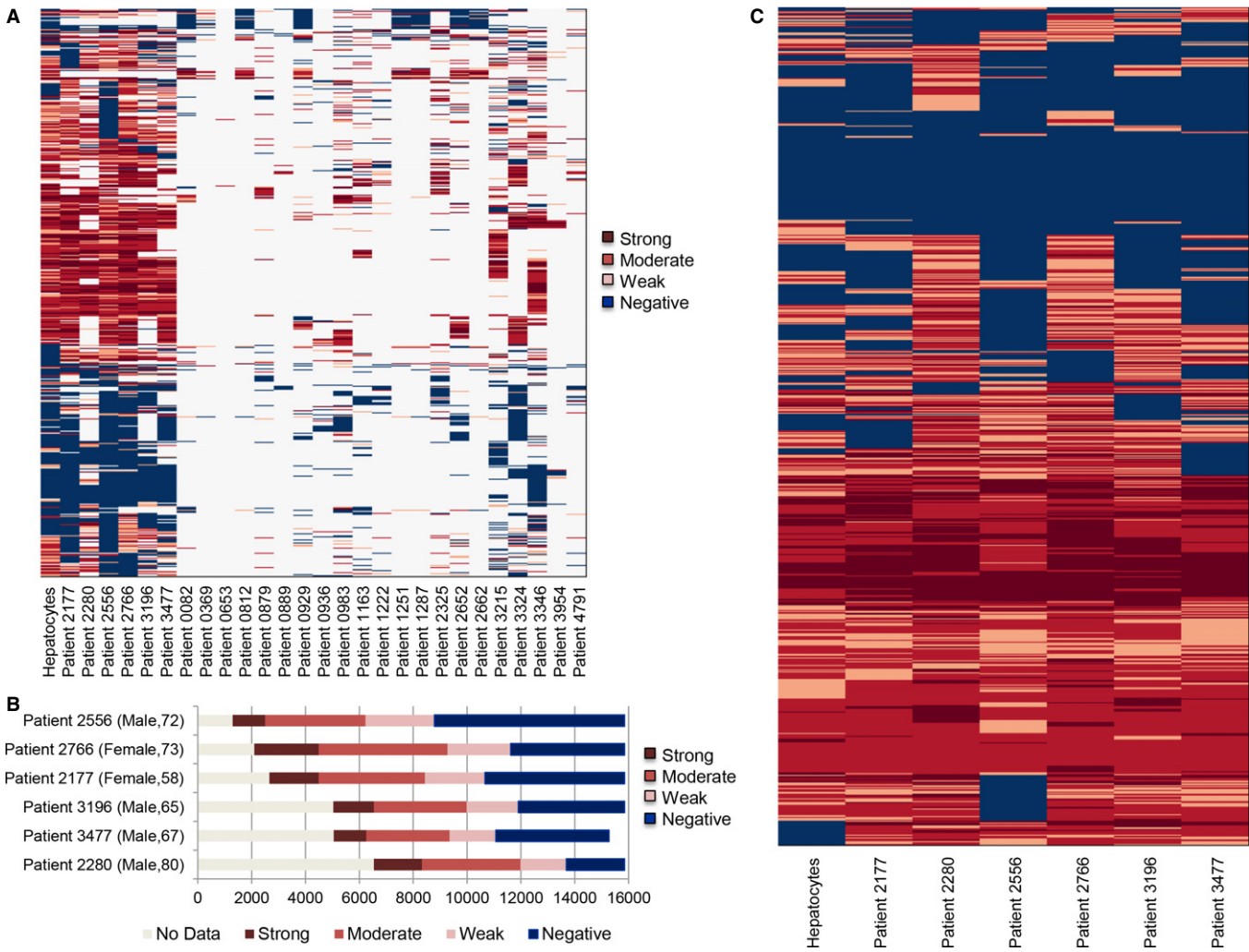

**Figure 2. Proteomics data for 27 HCC patients.**

A   Clustering of the generated proteomics data between 27 different HCC patients showed notable differences. The color indicates the protein expression differences between tissue samples.

B   Due to the coverage of the proteomics data, we focused on the reconstruction of the personalized models for six HCC patients. The number of the evaluated proteins in each HCC patients varies between 9,312 and 14,561.

C   The expression level of 4,936 proteins measured in all six HCC patients and the proteomics data showed notable differences between the HCC patients and hepatocytes.

Supplementary Dataset S7. These 56 metabolic tasks were categorized as energy and redox, internal conversions, substrate utilization and biosynthesis of products. Examples include the catabolism of nutrients or biosynthesis of precursors, but also generation of membrane potentials and ADP re-phosphorylation. For cancer cells, biomass growth was also included as an extra metabolic task. The objective was then to reconstruct models that can perform all the defined tasks and at the same time are consistent with the proteomics data. This concept was formulated in the tINIT algorithm (see Materials and Methods) and implemented in the RAVEN Toolbox (Agren *et al*, 2013).

Personalized GEMs for the HCC patients are provided in SBML format at the Human Metabolic Atlas (HMA) portal (www.metabol-icatlas.org), and the contents of the models are presented

(Table 1). The resulting personalized models ranged in size from 4,690 to 4,967 reactions and 1,715 to 2,025 genes. A total of 5,405 reactions and 2,361 genes were shared across the models and 4,212 of the reactions and 1,324 of the genes were present in all six personalized HCC models. It was observed that 248 of the reactions (Supplementary Fig S3A), 102 of the metabolites (Supplementary Fig S3B), and 227 of the genes (Supplementary Fig S3C) were unique to any one model. However, we observed notable differences between the reactions (Fig 3A) and genes (Fig 3B) during a pairwise comparison of the models. The differences between the numbers of reactions in the personalized models varied between 356 and 610, whereas the similarities between the reactions varied between 4,437 and 4,699. On the other hand, we observed larger differences in the number of the genes incorporated into the

**Table 1.  The content of the HMR 2.0, generic HCC model, and personalized genome-scale metabolic models for six HCC patients**

|  | Reactions | Metabolites | Genes | Model-specific reactions | Model-specific metabolites | Model-specific genes |
|---|---|---|---|---|---|---|
| Generic human model | | | | | | |
| HMR 2.0 | 8181 | 6007 | 3765 | – | – | – |
| Generic HCC model | | | | | | |
| HCC average model | 4816 | 4346 | 1779 | – | – | – |
| Personalized models for HCC patients | | | | | | |
| Patient 2177 | 4823 | 4339 | 1823 | 29 | 9 | 21 |
| Patient 2280 | 4954 | 4446 | 2025 | 89 | 49 | 74 |
| Patient 2556 | 4690 | 4324 | 1715 | 24 | 7 | 25 |
| Patient 2766 | 4967 | 4418 | 2009 | 36 | 8 | 57 |
| Patient 3196 | 4764 | 4377 | 1860 | 39 | 19 | 26 |
| Patient 3477 | 4833 | 4341 | 1803 | 31 | 10 | 24 |
| Common in all models | 4212 | 4113 | 1324 | – | – | – |
| Total shared components | 5405 | 4599 | 2361 | – | – | – |

models. The model differences on the present genes varied between 392 and 524, and considering all 2,361 genes shared in all models, a 16–22% difference was observed between the personalized models.

To investigate to which extent the personalized HCC models differ from a HCC population model, we reconstructed a generic HCC model based on the average protein expression of the 27 HCC patients and presented the content of the generic HCC model in Table 1. The pairwise reaction and gene differences between the personalized models and generic HCC model are also presented in Fig 3A and B, respectively. As it can be seen, the generic HCC model is about as different to the personalized models as these models are to one another.

In order to check whether a specific pathway is activated in the personalized and generic HCC models, we counted the number of the reactions in the relevant subsystem of HMR 2.0 (Supplementary Dataset S8). The numbers of the reactions in the personalized and generic HCC models did not show any significant differences. We further observed that none of the specific pathways were activated or deactivated in each of the personalized and generic HCC models. However, in-depth analysis showed that many reactions in the specific pathways differed between the models, and hence, this

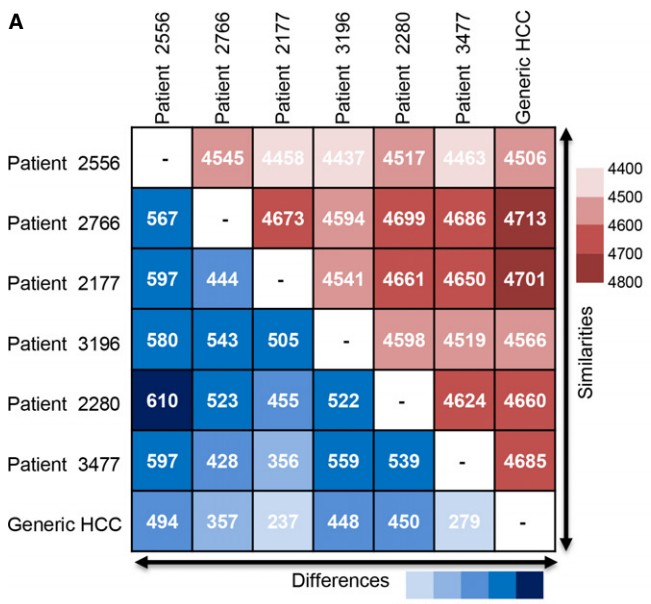
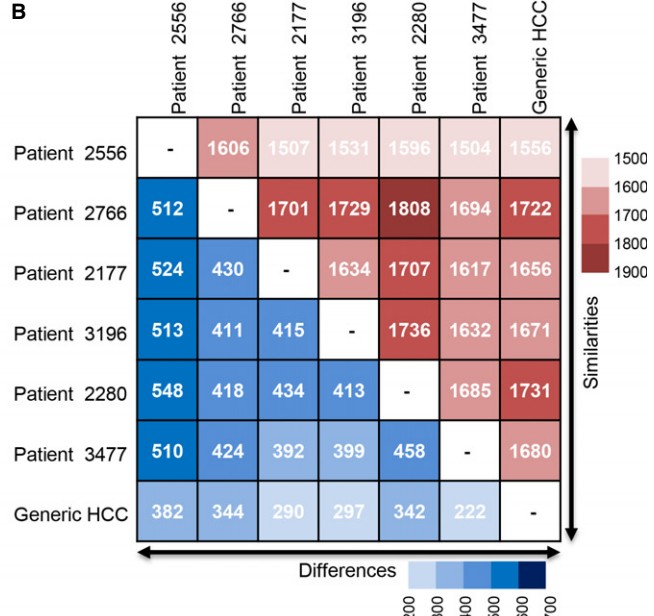

**Figure 3.  Comparison of the personalized GEMs for HCC patients.**

A, B   The pairwise differences and similarities of the reactions (A) and genes (B) between personalized HCC models and the generic HCC model that is reconstructed based on the average protein expression level of 27 HCC patients.

shows that different enzymes are activated in the different cancers but the same overall metabolic functions are characteristic of HCC.

## Cell type-specific functional GEMs

We reconstructed functional GEMs for 83 healthy cell types based on HMR 2.0 using tINIT algorithm. The proteomics data in HPA version 12.0, as well as the list of 56 metabolic functions known to occur in all cell types, were used as input to tINIT to generate functional cell type GEMs. The GEMs are available in SBML format at the HMA portal (www.metabolicatlas.org). The number of the reactions, metabolites, and genes in the models are presented in Supplementary Dataset S9. These models represent a significant improvement over existing automatically reconstructed cell type-specific models in that they are based on high-quality proteomics data and are built to carry out a range of important metabolic functions. This makes them an important resource to perform simulations or for the integration of omics data.

## Antimetabolites for HCC patients

Previously, Kim *et al* (2011) used the concept of metabolite essentiality to identify potential drug targets using a GEM for an opportunistic microbial pathogen. The targets were then used to identify novel potential antimicrobial targets from a chemical library, and a screening identified one chemical to have clear antimicrobial properties. The concept of metabolite essentiality is a similar approach to that of antimetabolites. An antimetabolite is structurally similar to a specific metabolite, but it cannot be utilized for the production of physiologically important substances. Its drug action comes from that it inhibits or otherwise affects the enzyme(s) that utilizes the metabolite. A strong characteristic of antimetabolites is that they can allow for targeting of multiple enzymes simultaneously and can reduce or kill the growth of tumors more effectively (Garg *et al*, 2010). Antimetabolites is one of the most widely used type of drugs for cancer treatment as of today (Hebar *et al*, 2013). Since antimetabolites accomplish their drug action from being structurally similar to metabolites, here we emulate the effect of antimetabolites by inhibiting enzymes based on which metabolites they have as substrates. We identified the metabolites that, upon blockage, disabled tumor growth in HCC patients while having a minimal effect on the previously defined biological processes of healthy cells (Fig 1C) (see Materials and Methods). The structural analogs of these metabolites were predicted as potential anticancer drugs and proposed as antimetabolites. Such analogs can be screened from chemical libraries in a similar manner to what was successfully used by Kim *et al* (2011) for the identification of novel antibiotics.

By means of our approach, we predicted 147 antimetabolites that can inhibit growth in any of the studied six HCC tumors (Supplementary Dataset S10). One hundred and one (69%) of these potential antimetabolites were predicted to be effective in disabling growth in all six HCC patients (Fig 4A), 23 (16%) of the antimetabolites were effective in 2–5 patients, and the remaining 23 (16%) of the antimetabolites were only effective in one of the patients (Fig 4B). The 46 (31%) of the antimetabolites that are predicted to be effective in only some of the HCC patients are presented in Fig 4C. Even though 69% of antimetabolites disable growth in all of the HCC tumors, the fact that not all antimetabolites are applicable for all the HCC tumors illustrates the importance of using personalized models rather than relying on a generic HCC model.

One hundred and one potential antimetabolites that were predicted to be effective in all six HCC patients were grouped based on the relevant subsystem in HMR 2.0 and literature evidence for their usage as antimetabolites in different cancers was included (Supplementary Dataset S11). The evidence for each metabolite was provided for HCC and for other cancers if it has not been used in the treatment of HCC yet. The number of antimetabolites in each subsystem is presented in Fig 5. It was observed that cholesterol biosynthesis contained the largest number of metabolites, followed by synthesis of co-factors, nucleotides, lipids, amino acids, and folate metabolism.

The analogs of 22 metabolites in folate, vitamin B12, and nucleotide metabolism were proposed as antimetabolites for inhibiting or killing the growth of tumors in all HCC patients. Vitamin B12 and folate metabolism may induce abnormal DNA methylation and DNA synthesis and the development of HCC is associated with an increased DNA synthesis (Leong & Leong, 2005). Fourteen of these metabolite analogs have already been used as antimetabolites in the treatment of different types of cancer (Supplementary Dataset S11) and the remaining eight metabolites were predicted to be beneficial for inhibiting the growth since the enzymes utilizing these metabolites have already been targeted in different cancers.

The analogs of 30 metabolites were targeting enzymes in the cholesterol biosynthesis and mevalonate pathways. Cholesterol is an essential molecule for building cell membranes and a precursor to several essential hormones and bile acids. The growth of HCC tumors is dependent on cholesterol biosynthesis (Borena *et al*, 2012), and use of statins that inhibit the enzyme HMG-CoA reductase has previously been proposed in the treatment of HCC (Lonardo & Loria, 2012). Although HMG-CoA reductase is the rate-limiting and key regulatory enzyme in the cholesterol synthesis, several other enzymes in cholesterol biosynthesis have been targeted for preventing the proliferation of the cells in different cancers (Cuthbert & Lipsky, 1997) (Supplementary Dataset S11).

The analogs of 13 metabolites in terpenoid backbone biosynthesis, two metabolites in pantothenate and CoA biosynthesis, one metabolite in nicotinate and nicotinamide metabolism, two metabolites in riboflavin metabolism, nine metabolites in lipid metabolism and eight metabolites in amino acid metabolism were identified as potential antimetabolites.

Two metabolites involved in the synthesis of tetrahydrobiopterin (BH4), which is an essential cofactor for several aromatic amino acid hydroxylases including tyrosine and tryptophan, were predicted as antimetabolites. BH4 is synthesized from GTP and supplementation of the BH4 has been shown to increase cell proliferation and effect tumor angiogenesis in endothelial cells (Chen *et al*, 2010). One of these metabolite analogs has already been used as antimetabolite. Tyrosine was also among the proposed antimetabolites for HCC, and tyrosine kinase inhibitors have been proposed as potential targets for the treatment of HCC (Giannelli *et al*, 2007).

The analogs of eight metabolites were identified in heme and porphyrin metabolism. The analog of 5-aminolevulinate, which can be synthesized from glycine and succinyl CoA and is a precursor for the synthesis of porphyrin, heme, and bile pigments (Ishizuka *et al*, 2011), has already been used as an anticancer drug. The correlation between

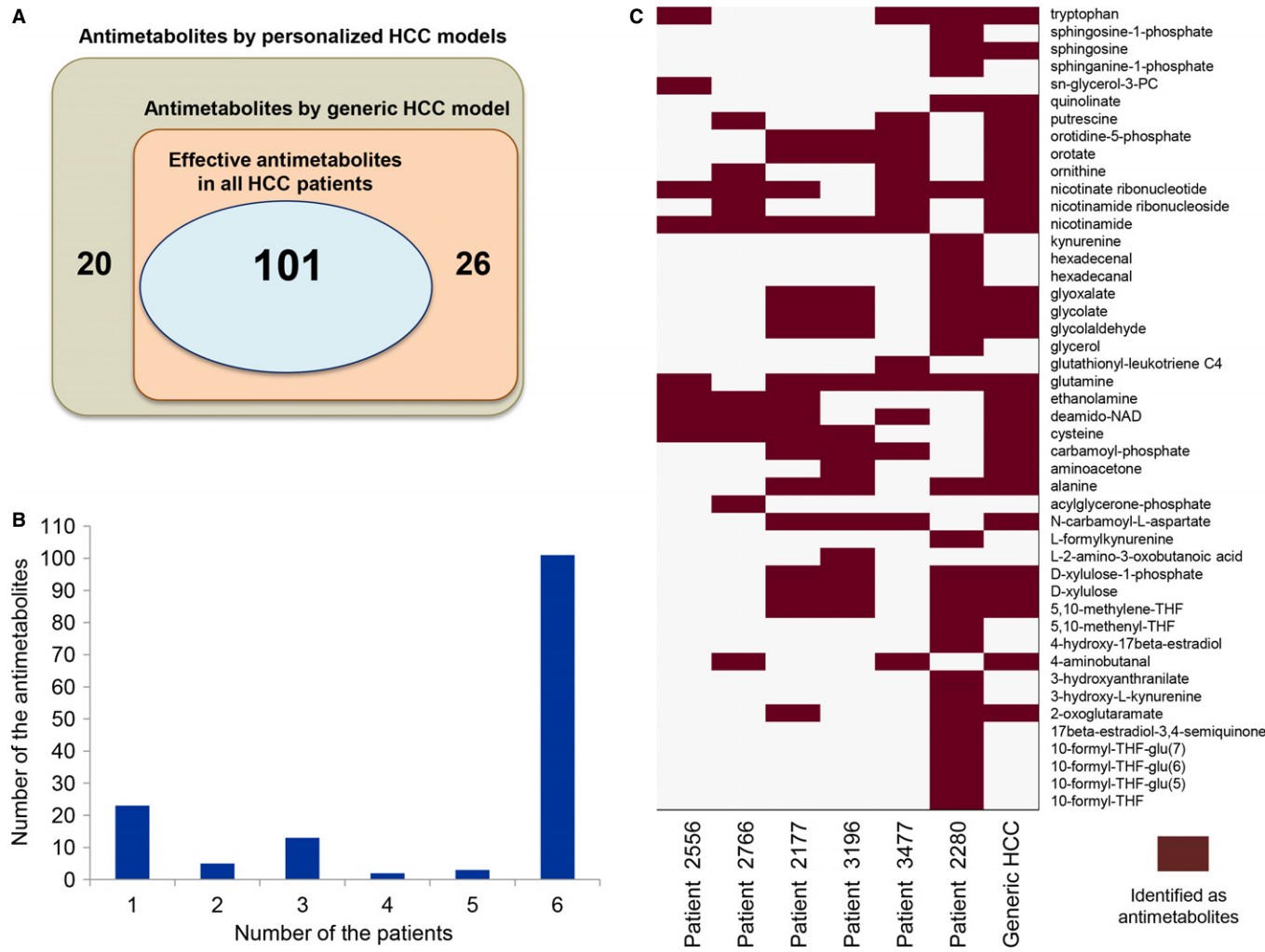

**Figure 4.   Prediction of antimetabolites for HCC patients.**

A   147 antimetabolites are predicted as potential anticancer drugs through personalized HCC models and 101 of these antimetabolites are effective for inhibiting HCC tumor growth in all six HCC patients. Antimetabolites are also predicted through the use of a generic HCC model that is reconstructed based on the average protein expression data in HCC patients, and 127 potential antimetabolites are identified. Twenty-six of the antimetabolites predicted based on the generic HCC models are not effective in all six HCC patients.

B   Distribution of the antimetabolites that are predicted to be effective in number of the personalized HCC models.

C   46 of the antimetabolites identified through the use of personalized models cannot be used for inhibition of the HCC tumor in all six HCC patients. The differences between the 46 predicted antimetabolites are shown through the use of personalized and generic HCC models.

activity of the remaining seven metabolites and the cancer progression was obtained in different cancers (Supplementary Dataset S11).

In summary, 22 of these 101 potential antimetabolites are in current use as anticancer drug targets and 61 of them have either been proposed as drug targets or show a strong correlation with cancer progression. Eighteen of targets have not been previously suggested as anticancer drugs. We also evaluated the toxic effect of these potential antimetabolites using 83 GEMs for healthy cells for testing the disruption of the antimetabolites on the healthy cell types in human body.

We also predicted antimetabolites through the use of the generic HCC model which was reconstructed based on the average HCC population data. Analogs of 127 metabolites were predicted as antimetabolites through the generic HCC model, and 26 (20%) of these

antimetabolites were not suitable for inhibiting the growth in all six HCC patients (Fig 4A). These 26 antimetabolites would therefore not be suitable targets for cancer treatment in all HCC patients. Thus, personalized HCC models allowed us to predict the effect of these false-positive antimetabolites on the treatment of all HCC patients. It is hereby clear that the use of personalized models can significantly improve the identification of drug targets effective in a given patient.

**The potential use of L-carnitine analog in the treatment of HCC**

The analogs of L-carnitine and metabolites involved in the synthesis of L-carnitine were proposed as antimetabolites for the treatment of all six HCC patients due to the non-toxic effect to here studied healthy cell types (Fig 6). L-Carnitine is synthesized in the liver and

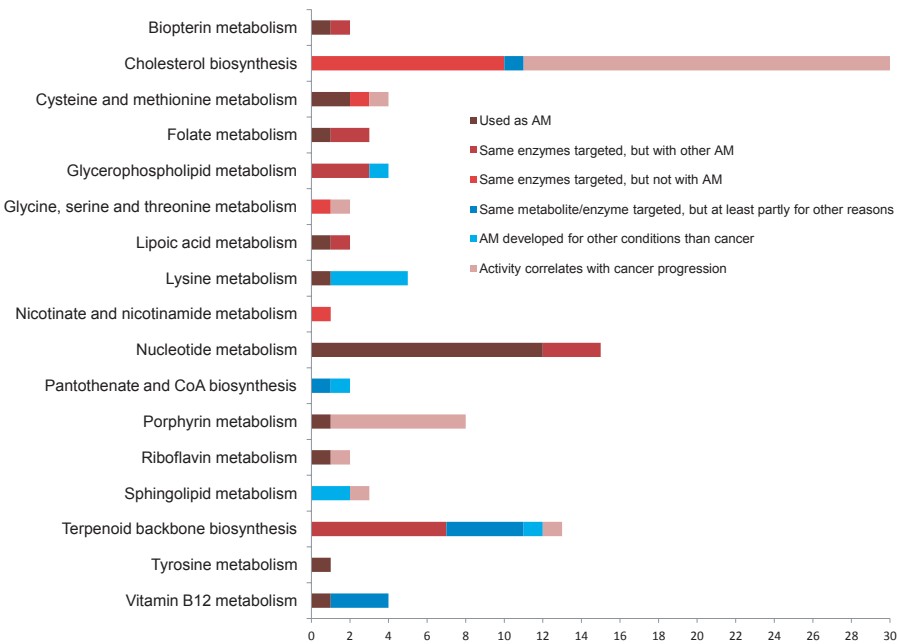

**Figure 5. Evidence levels of the predicted antimetabolites for HCC patients.**
Antimetabolites are a type of drugs, which acts by inhibiting the use of a normal metabolite, most often by being structurally similar to the metabolite in question. The 101 predicted antimetabolites were categorized based on their known use as antimetabolites and/or their connection to HCC (see Supplementary Dataset S11 for literature evidence). In summary, 22 of the predicted antimetabolites are currently in use as anticancer drugs, and 9 are used as drugs against other diseases. For 39 of them, the corresponding enzymes are currently targets for drugs but not with antimetabolites. The remaining 31 have not been investigated as drugs or drug targets, but all of them show a strong correlation with cancer progression. These results speak strongly for the validity of the *in silico* predictions.

kidneys, from the essential amino acids lysine and methionine, but it can also be derived from dietary sources (Rebouche, 1991). Notably, the analogs of lysine and methionine were also predicted as antimetabolites through our analysis and have already been used as antimetabolites in the treatment of different cancers. We also observed that proteins involved in the synthesis of the L-carnitine have strong or moderate expression levels in HCC tumors.

L-Carnitine has powerful antioxidant and anti-inflammatory properties and has a decisive role in the metabolism of FAs and energy by regulating the free CoA and acyl-CoA ratio in the mitochondria (Silverio *et al*, 2011). L-Carnitine is involved in the transport of activated long-chain FAs from the cytosol to the mitochondria where these mobilized FAs can be degraded through β-oxidation. FAs represent a very relevant energy source for many cells, including cancers, and can be taken up from outside the cell, synthesized through *de novo* synthesis or obtained through lipolysis in the liver.

L-Carnitine also facilitates the transfer of peroxisomal β-oxidation products to the mitochondria and removal of medium-chain FAs. Notably, our GO BP term enrichment analysis of hepatocytes and HCC tumor proteomics data suggested that the expression level of the genes involved in FA and lipid oxidation in HCC patients was higher than in healthy hepatocytes. Higher protein expression level of genes involved in β-oxidation can be explained by the increased functional activity of mitochondria in HCC. Previously, Wu *et al* (1984) measured the size, number, and surface membranes of the mitochondria in HCC tumor and reported a significantly higher activity of mitochondria. Furthermore, Toshima *et al* (2013) also reported the activation of the β-oxidation in HCC progression. The

β-oxidation process provides ATP and also supplies and converts nutrients in the liver (Carracedo *et al*, 2013).

In order to test the use of an L-carnitine analog as a potential antimetabolite for the inhibition of HCC tumor growth, we studied the effect of perhexiline, an inhibitor of carnitine palmitoyltransferase 1 (CPT1) and to a lesser extent CPT2, on the proliferation of the HepG2 cell line. In our study, we used perhexiline to mimic the effect of the L-carnitine analog since L-carnitine conjugates to FAs and translocates them to the mitochondria through the enzyme CPT1. We treated the HepG2 cells with four different concentrations (2, 4, 8 and 20 μM) of perhexiline, determined the viable cells after 24 and 48 h, and compared the inhibitory effect of perhexiline to sorafenib (2 and 4 μM) (Fig 7). Whereas the lower concentration (2 μM) had no effect, the results clearly showed that the treatment of the HepG2 cell line with 8 and 20 μM perhexiline reduced the viability of the HepG2 cell line after 24 and 48 h (Fig 7A) and that its effect is comparable to the effect of sorafenib (Fig 7B). Figure 7C shows HepG2 cells after 24 h of treatment with 20 μM perhexiline.

## Discussion

In order to reduce the public health burden of HCC, continued efforts are needed to identify novel drug targets for developing effective HCC treatment strategies. It is well known that cancer cells modify their metabolism in order to meet the requirements of cellular proliferation. The potential use of metabolic enzymes as therapeutic targets has led to a renewed interest in understanding

 

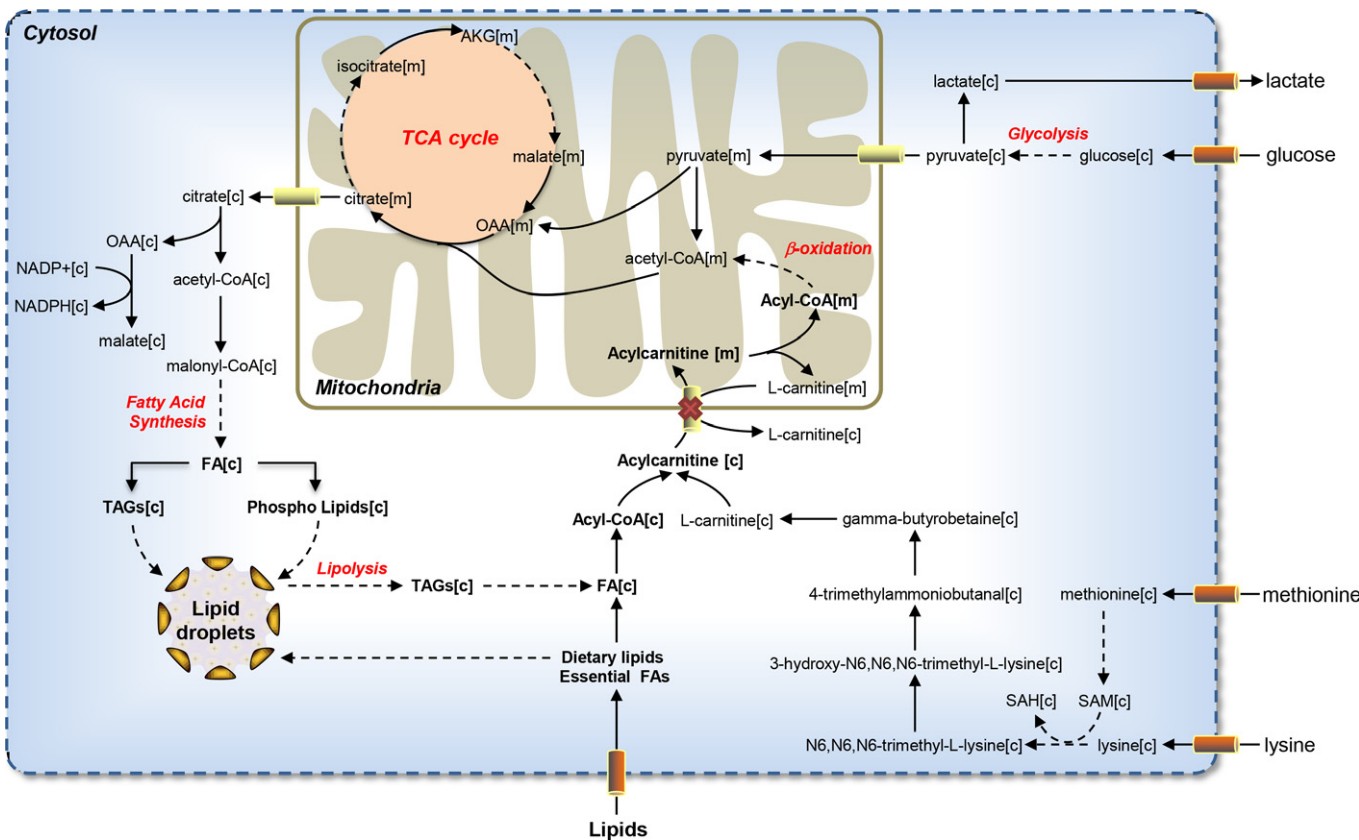

**Figure 6.  Usage of ʟ-carnitine antimetabolites for the treatment of HCC.**
ʟ-Carnitine and metabolites in the ʟ-carnitine biosynthetic pathway, as well as two essential amino acids, lysine and methionine, necessary for the synthesis of ʟ-carnitine were identified through our modeling approach. The analogs of ʟ-carnitine were proposed as antimetabolites for the treatment of HCC patients and the predicted consequence of the use of an ʟ-carnitine antimetabolite is presented. ʟ-Carnitine antimetabolites may result in reduced β-oxidation, *de novo* synthesis of fatty acids, and eventually may suppress or kill the growth of the HCC tumor. The abbreviations and the detailed explanations for the metabolites as well as the associated genes for each reaction are presented in HMR 2.0.

the altered metabolism in cancer (Vander Heiden, 2011). Our study showed a novel approach to identify therapeutic targets for treating HCC through combination of personalized proteomics data and metabolic modeling. This is the first time personalized GEMs have been used to find and evaluate new potential drugs, some that could be used for general treatment of HCC, and others that are highly specific for each HCC patient.

We generated high-quality personalized proteomics data for 27 HCC patients in order to understand the differences in tumor appearance and to search for potential drugs that would be effective in all patients. We developed the tINIT algorithm and reconstructed personalized GEMs for six HCC patients based on the personalized proteomics data and HMR 2.0. The tINIT algorithm was developed for the efficient reconstruction of simulation-ready models based on omics data coupled with defined metabolic tasks.

In a similar fashion, we also reconstructed functional cell type-specific GEMs for 83 different healthy cell types, which represent an important resource in their own. These functional GEMs may enable the application of constraint-based modeling techniques to distinguish metabolic states under different physiological conditions. This is a significant improvement over the previous automatically

reconstructed GEMs, since these functional models were generated based on high-quality proteomics data and in a manner that enables them to perform a range of defined biological tasks. These tasks can of course be expanded on, and functional models for other cell types can be reconstructed with the tINIT algorithm as implemented in the RAVEN Toolbox (Agren *et al*, 2013).

By using the concept of antimetabolites, we were able to propose anticancer drugs which could be effective in inhibiting tumor growth. Furthermore, we simulated the effect of these antimetabolites on 83 healthy cell types to predict their toxic effects. We thus identified 101 antimetabolites which were predicted to inhibit cancer growth in all six HCC patients simultaneously, while not being overly toxic to healthy cells. Through our personalized modeling approach, we also predicted 46 antimetabolites which would inhibit HCC tumor growth only in a subset of the patients. Therefore, the outcome of our study can be used to predict false-positive drug targets that would not be effective in all patients.

One of the few identified antimetabolites that have not yet been studied for cancer patients is antimetabolites of ʟ-carnitine. We predicted that such antimetabolites may inhibit the β-oxidation and hereby suppress the growth of HCC tumor (Fig 6). This was tested

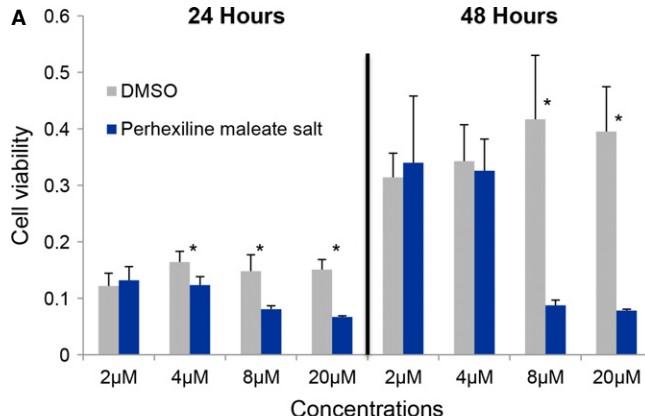

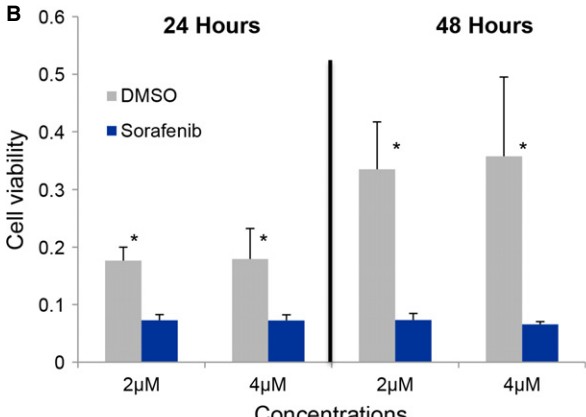

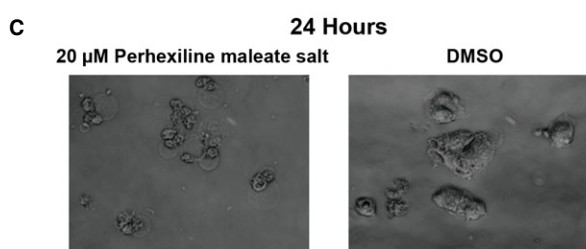

**Figure 7. Inhibitory effect of perhexiline on the proliferation of the HepG2 cell line.**

A, B    Perhexiline was used to mimic the effect of the L-carnitine analog on the proliferation of the HepG2 cell line. The number of viable cells was determined after treatment with (A) perhexiline (2, 4, 8, and 20 μM) and (B) sorafenib (2 and 4 μM) for 24 and 48 h. Both perhexiline and sorafenib were dissolved in DMSO, and for each concentration of compounds analyzed, controls with corresponding concentration of DMSO were analyzed. Each bar represents the results from eight replicate samples, and mean ± s.d. values are presented. Students t-test versus untreated cells: *P-values < 0.001.

C    Example images for the HepG2 cell line after 24 h of the treatment with 20 μM perhexiline and corresponding concentration of DMSO.

*in vitro* using perhexiline, which has been used to mimic the behavior of an L-carnitine analog, and it was shown to reduce the viability of HepG2 cells. This is a proof of principle that genome-scale modeling can be used to predict relevant targets for anticancer drug identification.

Several other studies have also proven the link between β-oxidation and different types of cancer and support our here presented results. β-Oxidation is a central pathway for energy generation in prostate cancer (Liu, 2006), and enhanced mitochondrial β-oxidation has been associated with tumor promotion in pancreatic cancer (Khasawneh *et al*, 2009). Inhibition of β-oxidation has been shown to induce apoptosis in leukemia cells and in glioblastoma cells (Samudio *et al*, 2010; Pike *et al*, 2011). It is earlier reported that expression of CPT1C, a brain-specific enzyme, in cancer cells promoted FA oxidation, ATP production, resistance to glucose deprivation, and tumor growth (Zaugg *et al*, 2011). The expression of CPT1C, which is frequently up-regulated in human lung tumors, increased the resistance to the mTOR complex 1 (mTORC1) inhibitors and was therefore proposed as a new therapeutic target for the treatment of hypoxic tumors. Recently, Pacilli *et al* (2013) reported that the inhibition of the CPT1A *in vivo* models of Burkitt's lymphoma induced lipid accumulation in cytosol and reduced the availability of cytosolic acetyl-CoA. It should also be noted that the level of acetyl-CoA in a cell regulates the oxidation of fatty acids and pyruvate (Pekala *et al*, 2011) and effects the activity of pyruvate dehydrogenase (PDH) complex that provides the link between glycolysis and the TCA cycle. The usage of L-carnitine antimetabolites, and eventual inhibition of the β-oxidation, may also result in a decrease in cytosolic NADPH production. Decreased level of NADPH in the cytosol results in increased production of reactive oxygen species resulting in cell death within HCC tumor as in glioma cells (Pike *et al*, 2011).

In conclusion, we identified potential antimetabolites which may inhibit the growth of HCC tumors through the use of personalized metabolic modeling, proposed the usage of antimetabolites for the treatment of HCC, and demonstrated the inhibitory effect of the L-carnitine analog, one of the predicted antimetabolites, on the proliferation of the HepG2 cell line. The results of our study can be used to reduce the number of chemical compounds for drug screening by focusing on the structural analogs of the identified antimetabolites. It is of course important to note that the findings presented here are based on cellular models and do not take systemic effects into consideration. One way to alleviate this could be to integrate the method described here with whole-body pharmacokinetic and pharmacodynamics modeling. Beyond the prediction of new potential drug targets for HCC, the modeling approach presented here may be expanded to predict the effect of a standard therapy for a particular individual and evaluate whether the treatment is likely to work. Our approach may hereby enable new exciting possibilities for personalized medicine.

# Materials and Methods

### Proteomics data for HCC patients

The HPA portal covers the relative abundance of proteins analyzed with one or more antibodies and the subcellular localization of proteins in all major human healthy cells and cancer (Uhlen *et al*, 2010). The proteomic profiling using immunohistochemistry was performed as previously described (Uhlen *et al*, 2005). In brief, tissue microarrays (TMAs) were produced for HCC tumors obtained from 27 different HCC patients. Representative formalin- and

paraffin-embedded material from donor blocks were punched (1 mm in diameter) and placed in a recipient block TMAs (Kampf *et al*, 2012). Thereafter, 4-μm TMA sections were cut using a microtome and placed on super frost glass slides.

Immunohistochemically stained tissues were scanned and digitalized at 20× magnification. Annotations of each high-resolution image was manually performed by certified pathologists with the relative expression levels including strong, moderate, weak, and negative (Kampf *et al*, 2004), and proteins with strong, moderate, and weak relative expression were included in the reconstruction process of the GEMs. The annotation of the presence or absence of a particular protein target as well as related high-resolution images is publically available through the Human Protein Atlas (www.proteinatlas.org). During the reconstruction process of personalized GEM for six HCC patients, the missing expression of protein in each patient was predicted using the median of the protein expression levels in other 26 HCC patients. The present and absent proteins in each patient which were input for tINIT are presented in Supplementary Dataset S5.

### Task-driven reconstruction using tINIT

Previously, Agren *et al* (2012) developed the INIT (Integrative Network Inference for Tissues) algorithm for automated generation of cell type-specific and cancer GEMs. These networks could be seen as snapshots of active metabolism in a given cell type, but they were not functional models that could be directly applied for simulations. In this work, we expanded significantly the INIT algorithm in order to allow for direct reconstruction of functional GEMs. The tINIT algorithm allows the user to define metabolic tasks, which the resulting model should be able to perform. These metabolic tasks can be outlined and used as an input to tINIT algorithm in Microsoft Excel for convenience (Supplementary Dataset S7).

The algorithm then works by first identifying the set of reactions in the generic model which, if any of them are excluded, cause one or more of the tasks to fail. This set of reactions then have to be in the resulting model. Note that this is not the same thing as the smallest set required for performing the tasks, as there can be isoenzymes or alternative pathways. The INIT algorithm is then implemented as described in the original paper (Agren *et al*, 2012), but with the additional constraint that these reactions have to be in the solution. The resulting solution has to adjust to fit with these reactions and is therefore likely to be close to being able to perform the tasks. In a final step, each task is tested in a sequential manner, and if it cannot be performed, then the gap-filling algorithm in the RAVEN Toolbox (Agren *et al*, 2013) is applied in order to enable it. The sequential testing means that the order of the tasks could theoretically impact which reactions are included in the gap-filling step. However, this is normally not the case, since the solution is close to functional because of the set of required reactions.

tINIT contains two additional improvements over the original INIT algorithm. Firstly, it constrains the solution so that reversible reactions cannot have flux in both directions simultaneously. This enabled some loops to be included even though they were not connected to the rest of the metabolic network. Secondly, it allows the user the choice of whether net production of all metabolites should be allowed (which was the case in the original implementa-

tion). The tINIT algorithm is implemented and extensively commented in the RAVEN Toolbox (Agren *et al*, 2013) together with functions for working with the concept of metabolic tasks (www.sysbio.se/BioMet).

The tasks used for imposing constraints on the functionality of the reconstructed models (see Supplementary Dataset S7) are based on metabolic functions that are known to occur in all cell types. As such, there is some redundancy between the tasks. For example, ADP re-phosphorylation is a prerequisite for the biosynthesis of some of the precursors (which in turn is a prerequisite for biomass formation). The reason for this is twofold. On the one hand, it makes for a less computationally demanding optimization, as the reconstruction can be performed in a more stepwise manner. On the other hand, it makes for more fine-grained analysis of the impact of each task, in particular when it comes to the effect of antimetabolites. It should be noted that the redundancy is not a problem from a reconstruction viewpoint, since the resulting GEM will look the same regardless.

Since the set of measured proteins differed somewhat between the six HCC patients, averaged data from all 27 HCC patients were used to fill the gaps. tINIT requires that all reactions in the reference network are given a score, and the alternative solution would be to use an arbitrary negative score for proteins which were not measured in some given patient. This would represent a larger bias, and it was therefore chosen that averaged data should be used for the missing protein expression value. It should be also noted that the filled protein expression data are relatively small comparing to the measured protein expressions in each patient.

### Prediction of antimetabolites and their toxicity

The effect of antimetabolites was emulated by considering the corresponding real metabolites as antimetabolite analogs. For each unique metabolite (not taking compartmentalization into account), all reactions that used it as a substrate were constrained to have no flux. Reversible reactions were constrained to have no flux in the direction where the metabolite was a substrate. The model was then tested for its ability to perform each of the 56 tasks defined in Supplementary Dataset S7 by using the checkTasks function in the RAVEN Toolbox. This was done for each of the six HCC models and for the 83 GEMs for healthy cell types.

In total, 3,160 metabolites were tested in this manner, out of which 410 disabled growth in at least one of the HCC models (Supplementary Dataset S10). 349 out of these disabled growth in all six HCC models and this led to a total of 349 potential antimetabolites. However, a large fraction of these were fatty acid derivatives involved in the formation of pool metabolites. These were deemed to be unlikely to be effective antimetabolites and were thus excluded. This resulted in a final list of 147 metabolites that are potential antimetabolite homologs. One hundred and one of these potential antimetabolites were effective in killing the growth of the tumors in all six HCC patients.

Antimetabolites can also be disruptive to healthy cells, similar to other chemotherapeutic drugs (Munoz-Pinedo *et al*, 2012). It is therefore a need for evaluating the toxic effect on health cells. We used the 83 GEMs for healthy cells as a means to investigate this. As a first step, metabolites which disabled tasks classified as being

involved in energy and redox balancing in any of the 83 GEMs for healthy cells were excluded. The rational for this was that such tasks could be supposed to be central for all types of cells, not only proliferating ones.

**Cell proliferation assay on liver cancer cell line HepG2**

Hepatocellular carcinoma cell line HepG2 was obtained from DSMZ (DSMZ, Braunschweig, Germany) and cultivated according to DSMZ instructions. A proliferation assay was performed using the colorimetric CellTiter 96 AQ$_{ueous}$ One Solution Cell Proliferation Assay (MTS) (Promega, Fitchburg, USA) according to instructions from the manufacturer. In brief, HepG2 was separately treated with 2 and 4 μM of sorafenib (Santa Cruz Biotechnology, Inc., Dallas, USA) and with 2, 4, 8, and 20 μM of perhexiline maleate salt (Sigma-Aldrich, St. Louis, USA) for 24 and 48 h. Both compounds were dissolved in DMSO, and corresponding concentrations of DMSO in the medium were used as controls. In addition, controls consisting of cells growing in only cell culture medium were included. For all concentrations of sorafenib and perhexiline maleate salt, and for corresponding DMSO controls, eight replicates were analyzed. For the controls consisting of cells in only medium, 16 replicates were analyzed. For all experiments, the measured colorimetric differences between DMSO controls and medium controls were insignificant.

**Supplementary information** for this article is available online: http://msb.embopress.org

## Acknowledgements

This work was supported by grants from the Knut and Alice Wallenberg Foundation.

## Author contributions

RA and AM developed the tINIT algorithm and method for identifying antimetabolites. AM and RA performed the analysis of clinical data. CK, AA, and MU coordinated the generation of the proteomics data and *in vitro* experiments. JN and MU conceived the project. AM, RA, and JN wrote the paper, and all authors were involved in editing the paper.

## Conflict of interest

The authors declare that they have no conflict of interest.

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
