## [Review Process File · Molecular Systems Biology]

Identification of anticancer drugs for hepatocellular carcinoma through personalized genome-scale metabolic modeling

Rasmus Agren, Adil Mardinoglu, Anna Asplund, Caroline Kampf, Mathias Uhlen, Jens Nielsen

Corresponding author: Jens Nielsen, Chalmers University of Technology

Review timeline:

Submission date:	24 July 2013
Editorial Decision:	12 September 2013
Appeal:	07 October 2013
Editorial Decision:	18 October 2013
Re-submission:	31 October 2013
Editorial Decision:	18 November 2013
Re-submission:	14 January 2014
Editorial Decision:	17 February 2014
Revision received:	18 February 2014
Accepted:	20 February 2014

Editor: Maria Polychronidou

Transaction Report:

1st Editorial Decision

12 September 2013

Thank you again for submitting your work to Molecular Systems Biology. We have now heard back from the three referees whom we asked to evaluate your manuscript. As you will see from the reports below, the referees raise substantial concerns on your work, which, I am afraid to say, preclude its publication in Molecular Systems Biology.

Overall, the reviewers acknowledge that the topic of the study is potentially interesting. However, they raise significant concerns with regard to the impact of the main findings and the conclusiveness of the study. In particular, the reviewers point out that the superiority of the presented approach over existing methods as well as its applicability for drug identification remain to be demonstrated. Moreover, the reviewers refer to the lack of experimental validation of the presented predictions. As such, I am afraid that the reviewers provided only a very limited level of support.

Under these circumstances, I see no other choice than to return the manuscript with the message that we cannot offer to publish it. In any case, thank you for the opportunity to examine your work. I hope that the points raised in the reports will prove useful to you and that you will not be discouraged from submitting future work to Molecular Systems Biology.

REFREREE REPORTS

Reviewer #1:

Agren et al use a detailed data set of protein expression (reduced to the binary case) of 27 hepatocellular carcinoma isolated from patients. For 6 samples, they generated a minimal metabolic network by pruning the complete HMR 2.0 model subject to the boundaries imposed by the expression data and a set of predefined metabolic capabilities that the minimal network must fulfill. The approach builds on a previous method by the same group and is named tINIT.

In order to identify compounds to efficiently and selectively treat HCC, the query the metabolic reconstructions for metabolites whose removal impairs fulfillment of the predefined metabolic capabilities in tumors but not in reconstruction of healthy cell types. For this they postulate that for every metabolite present in the network, it is possible to generate a variant (antimetabolite) which is metabolically inert and capable of virtually silencing all enzymes that use the metabolite as substrate. About hundred metabolites in a dozen pathways could be proposed. The predictions are discussed at large with literature data, but no experimental validation presented.

Major comments:

(1) A pillar of the manuscript is the development of the tINIT method for the reconstruction. Albeit it constitutes a minor improvement of a previous method, I like the idea. However, the manuscript fails to test its merits. I had a cursory glance at the tasks in the supplement to understand. My impression is that the list is arbitrary and also redundant. A few examples: Why are cells supposed to consume non-essential amino acids? Why are cells assumed to synthesize intermediary compounds (e.g. glycolytic compounds) while these are anyway necessary for the task of growing? Why are cells obliged to grow on HAM's medium while they were sampled from biopsies? What is the sense of constraining mitochondrial acetyl-CoA de novo synthesis by the "acetyl-CoA[m] => CoA[m]" reaction?

To appreciate the novelty of the method and its benefits over previous, the authors should (1) explain the rationale of the different tasks; (2) compare the results obtained with and without tasks (tINIT vs INIT); (3) systematically compare the impact and independency of single tasks.

(2) The second cornerstone of the manuscript is the personalized genome-scale modeling. Important claims are made, but there are several flaws. First of all, the reconstruction is based on population-averaged data. It is not clear to me why so many missing values are present or what the cause for this is. Nevertheless, using median data from the entire population to fill gaps is a killer. The authors should refrain from it, and use only the proteomic data obtained for the 6 patients with dense measurements. This is a precondition to claim that the reconstruction was made based on n=1. This is likely going to reduce the size of the reconstructed models, but better reflects the data that one would get from a single patient.

(3) On the same note, the authors sought for antimetabolites that are equally affecting all 6 HCC networks. This doesn't prove that it's important to use personalized models as hastily claimed in the results. For a fair comparison, the authors should perform the antimetabolite sensitivity analysis using the data averaged over all 27 HCCs and demonstrate that it leads to a substantially different list of targets. Notably, a diverging result would not confirm that personalized reconstructions are better than averaged ones because we lack any experimental validation to benchmark the prediction. The authors should be more critical on these aspects.

(4) The authors claim that the personalized, proteomics-based models "represent a significant improvement over existing...", but there is really no prove for it. In fact, I think that the potential value of such a resource is undersold. Differences are superficially discussed based on GO enrichments, but the peculiarities of the GEMs are hidden under the carpet. In the context of this work, I expect the authors to detail on what pathways are specifically activated or deactivated in each of the 6 HCC of the patients (reconstructed without population data), compared to the reconstruction obtained from averaging all 27 individuals and to hepatocytes. To make a case for personalized network reconstructions, the authors should then emphasize the occurrence of personalized pathway usage, which in turn can be linked to specific sensitivity of individual HCCs to antimetabolites or enzyme inhibition.

To sell the GEMs as a resource for the community, more emphasis should be given also to the 83 networks obtained from healthy tissues.

(5) I have a few comments on the comparison of antimetabolite inhibition in HCC vs healthy GEMs.

(1) Was it based on metabolites that are present in the intersection of all GEMs or their union? The former is correct, the latter causes several trivialities which should be complemented by sound statistics. (2) Among the 83 healthy controls, are there any cell types which are growing/proliferating comparably to a carcinoma?

(6)The potential of antimetabolites is overblown. It is misleading to state that "antimetabolites" were identified. The authors predicted only metabolites which are essential in HCCs but not in healthy cells. This is not equivalent to identifying antimetabolites, for instance as Kim et al did with experimental verification. Please tone down and correct the statements throughout the manuscript.

(7)It is not clear why antimetabolites should be preferred to specifically inactivating single enzymes. This is a much more intuitive and practicable approach in drug development. Given that the GEMs are available, a computational analysis should be added to compare the two approaches.

Reviewer #2 :

This is a very interesting paper focused on a very important problem, namely how to take advantage of high-throughput data characterizing cancer cell physiology. The authors have extended previous work on an algorithm for developing "personalized" models that integrates data to create functional, predictive computational models. Below are some concerns:

Many grammar issues throughout that must be corrected for clarity of the presentation; e.g., "therefore inferring with the corresponding enzymes" in the Summary...should be "interfering".

The difference in the # of reactions between the personalized models is surprisingly (to this reviewer) small. Is there any kind of indication about why the variation is at the level it is? What are the primary drivers of the inclusion/exclusion of reactions during the implementation of this tINIT algorithm? Are any of the 67 metabolic "tasks" more important than others for driving the inclusion/exclusion decisions?

Perhaps the authors can speculate on why there is such a small number of reactions unique to any one model. Is this an artifact of the models? Or perhaps there is some biological significance to that observation?

The validation of antimetabolites was all done retrospectively. While such a comparison provides some nice support for the model predictions, the paper should be a little more explicit in delineating the workflow (same database from which the protein expression was used to build the models apparently also contained data on which antimetabolites were effective). This doesn't demonstrate solid statistics for accuracy of the predictions (perhaps the database tested all of the other metabolites that were predicted to be effective but simply did not report).

6th paragraph of the antimetabolite section, last sentence, says that "identified as antimetabolites and these metabolites can provide effective treatment strategies"...should probably be "could" or "may" or something else similar. As stated it sounds more like these have been tested. Similar to point in previous paragraph, the authors need to be more explicit about what was predicted, tested, etc. so that claims are not over-stated.

The authors make the frequent claim that their models derived from proteomics data are of a higher quality than models that would have been developed from expression data. However, such a claim, while perhaps attractive, is not necessarily true. Just as one example, the quality of the antibodies for the proteomics data could be variable. Perhaps some simple comparison of the differences or value-added from model generation from the proteomics data could be of value and provide great confidence in the perhaps uniquely high quality of the predictions/outcomes reported here.

The comparison of the antimetabolite effects on the cancer cells versus their effects on the 83 healthy cell types was not fully developed. What might those results teach us? What were some of the patterns of antimetabolites that were predicted to be effective in cancer but for which there was a

potentially high deleterious effect in healthy cell types?

Reviewer #3 :

In their paper "Identification of anticancer drugs for hepatocellular carcinoma through personalized genome-scale metabolic modeling", Agren et al. describe the construction of personalized genome-scale metabolic models (GEM). In the following, the models are used to identify antimetabolites which may be used as anti-cancer drugs.

The paper by Agren et al. describes the next step in the emerging field of human GEMs. In particular the consideration of personalized models is a novel approach which may support the development of individualized treatment designs in the future. The presented work is hence of interest to the MSB readership. There are however, some ambiguities in the MS which need to be addressed by the authors before its final acceptance.

Comments:

ı The authors construct 6 GEMs for HCC patients and 83 cell-type specific GEMs. While the latter are only required to perform metabolic functions, the former are additionally tested whether they can additionally form biomass. This bilateral approach presents a fundamental concept in pharmacology to identify efficient (personalized GEMs) but non-toxic (cell-type specific GEMs) drugs. The authors should discuss their work within the context of efficacy vs. toxicity. Maybe they can even come up with a generic workflow outlining how these two aspects can be distinguished with different kinds of GEMs.

ı When constructing the GEMs, the authors probably need some cutoff criteria to distinguish strong, moderate and weak expression. Which cutoff was chosen? Is the structure of the GEMs robust for different value of the cutoff value?

ı In the last paragraph of the paragraph "Personalized GEMs for HCC patients" the authors discuss the statistics of the GEM construction. Please also include some biochemical/physiological discussion of the pathways which are unique. Is this an artifact of network construction (coverage) or an inherent property of the personalized network model?

ı The same questions arises in the next paragraph "Antimetabolites for HCC patients". Is the finding that some metabolized are only effective in certain patients due to personalized genome structure or due to network coverage during model construction.

ı Figure 1: The fact, that m11 is identified differently than m4 m12 should be visualized differently and not only by a dashed line.

ı The present paper largely covers pharmacodynamic effects of drug therapy. This, however, is only half of the story since pharmacokinetics are equally important for a targeted therapeutic design. GEMs have been integrated before in whole-body PBPK models to address amongst others PK/PD behavior or drug-induced intoxication (which is almost identical to the toxicity test in the manuscript). The authors should discuss their work within the context of this earlier study.

ı It is a frequent misconception to consider cellular models (and cellular assays) as an adequate surrogate for real patients. Though the authors avoid this impression in their work, they should nevertheless spend some time discussing the relevance (and limitations) of cellular models for real clinical applications.

ı In the same regard, the authors should discuss their GEMs within the context of inter-individual variability. Can variations in the 6 personalized GEMs be compared to inter-individual variability in real patients?

ĩ Finally, the authors should give some conceptual outlook: What is needed to translate findings in GEMs to the clinic?

ĩ Notably, all 6 HCC patients considered are older than 58 years. Is age a bias in the approach? Is more meta-information available regarding the background of the donors? The authors should discuss this.

Appeal

07 October 2013

On behalf of all the authors, I am writing to appeal your decision about our reviewed manuscript (MSB-13-4728) entitled "Identification of anticancer drugs for HCC through personalized genome-scale metabolic modeling".

We would like to thank all three reviewers for their comments and insights. We believe that these comments will only help us to further improve the description of the used methods and the presentation of the personalized proteomics data and genome-scale metabolic models. We are also pleased to see that all reviewers have found general interest in our manuscript and that none of the reviewers raise any major issue about the conclusions reached in the manuscript. In case of submitting the revised version of the manuscript, we can address the specific comments of the each referee by providing point-by-point response. Please see our attached response for each comment of the referees. Please also let us know if you want us to provide more detailed explanation in relation to any specific comment.

We are also confident that our revised paper will be of general interest to researchers in the field of metabolism, systems biology, systems medicine and metabolic network modelling, and therefore may be of interest for Molecular Systems Biology.

Reviewer #1:

Agren et al use a detailed data set of protein expression (reduced to the binary case) of 27 hepatocellular carcinoma isolated from patients. For 6 samples, they generated a minimal metabolic network by pruning the complete HMR 2.0 model subject to the boundaries imposed by the expression data and a set of predefined metabolic capabilities that the minimal network must fulfill. The approach builds on a previous method by the same group and is named tINIT.

In order to identify compounds to efficiently and selectively treat HCC, they query the metabolic reconstructions for metabolites whose removal impairs fulfillment of the predefined metabolic capabilities in tumors but not in reconstruction of healthy cell types. For this they postulate that for every metabolite present in the network, it is possible to generate a variant (antimetabolite) which is metabolically inert and capable of virtually silencing all enzymes that use the metabolite as substrate. About hundred metabolites in a dozen pathways could be proposed. The predictions are discussed at large with literature data, but no experimental validation presented.

We wish to thank Reviewer #1 for detailed reading of our manuscript and providing constructive comments. A small point is that the expression scores aren't reduced to binary and that the method is not about finding the minimal set of reactions which include the positive scoring reactions (this would be similar to the GIMME algorithm). Rather, the objective is to find a network which maximizes the sum of scores while being capable of performing all the metabolic tasks. This is a more flexible and complex task.

Major comments:

(1) A pillar of the manuscript is the development of the tINIT method for the reconstruction. Albeit it constitutes a minor improvement of a previous method, I like the idea. However, the manuscript fails to test its merits. I had a cursory glance at the tasks in the supplement to understand. My impression is that the list is arbitrary and also redundant. A few examples: Why are cells supposed to consume non-essential amino acids? Why are cells assumed to synthesize intermediary compounds (e.g. glycolytic compounds) while these are anyway necessary for the task of growing? Why are cells obliged to grow on HAM's medium while they were sampled from biopsies? What is the sense of

constraining mitochondrial acetyl-CoA de novo synthesis by the "acetyl-CoA[m] => CoA[m]" reaction?

To appreciate the novelty of the method and its benefits over previous, the authors should (1) explain the rationale of the different tasks; (2) compare the results obtained with and without tasks (tINIT vs INIT); (3) systematically compare the impact and independency of single tasks.

The rationale behind the metabolic tasks was to include all functions for which we had literature evidence that they can be performed by all cell types. It was a major oversight that the reference list wasn't present in the submitted Supplementary material, but this has been fixed now. The reason for having partly redundant tasks is twofold:

1) We wanted to have a comprehensive list of functions which all cells can perform in order to have a more fine grained view of how different antimetabolites affect the metabolic capabilities. An example would be in the point raised by the reviewer above, where all cells can synthesize glycolytic compounds but only the cancer cells must be able to use them for growth.

2) It is a computational advantage to split more complex task (such as growth) into its simpler constituents (such as biosynthesis of precursors) due to how tINIT is implemented. It also gives a greater understanding of which reactions are added for which tasks (rather than just add a bunch of reactions to enable growth in one step). Redundant tasks are no issue since it's the union of the essential reactions that are "forcefully" included in the network. Adding the same, or partly overlapping, tasks several will have no effect on the output.

The reason for using HAM's medium is that it is a well-defined medium that enables growth of many cell types. It has also been used for the same purpose by other groups when simulating cancer growth in vivo using GEMs. If we had used, say, metabolomics of blood or tissue instead the same question could be asked there.

De novo synthesis of mitochondrial acetyl-CoA is expressed like that because "de novo" refers here to the activated acetyl group rather than to the acetyl-CoA. The difference is that with a formulation like "acetyl-CoA[m] => acetyl-CoA[excreted]" the network would also need to have CoA synthesis. This is a separate task, and since it's a co-factor it isn't synthesized in any large amounts.

We fully agree regarding all the suggested additional work, and it is something that we are currently including in the paper. This work could be completed in a short amount of time. In particular, we think that a comparison between tINIT and INIT was missing, and that it will strengthen the paper considerably.

(2)The second cornerstone of the manuscript is the personalized genome-scale modeling. Important claims are made, but there are several flaws. First of all, the reconstruction is based on population-averaged data. It is not clear to me why so many missing values are present or what the cause for this is. Nevertheless, using median data from the entire population to fill gaps is a killer. The authors should refrain from it, and use only the proteomic data obtained for the 6 patients with dense measurements. This is a precondition to claim that the reconstruction was made based on n=1. This is likely going to reduce the size of the reconstructed models, but better reflects the data that one would get from a single patient.

We evaluated the presence/absence of proteins encoded by 15,841 genes in HCC tumors obtained from 27 patients using immunohistochemistry and presented personalized proteomics data. However, we reconstructed models only for the six HCC patients with the best coverage (between 9,312 and 14,561 genes measured). Also note that only a subset of these genes code for enzymes. This still constitutes a very large coverage compared to other published data sets.

We disagree with this comment of reviewer #1 about the usage of the average HCC data for the missing protein evidence. We discussed this internally during the preparation of manuscript and we think it is more accurate to use average data rather than using no evidence. This is a standard approach in many bioinformatics applications, and could be viewed as making the most of the available data. The fact that some small proportion of the measurements was based on population does not render the models "non-personalized". Also, the way that tINIT is designed; it would not be

a good solution to simply reconstruct models without the population data, as it would rely on some arbitrarily chosen value instead.

Another important reason is that we use model differences during the identification of the antimetabolites, and it would be a bigger bias to compare models for which different sets of proteins had been measured. We agree with the reviewer in that this is not an ideal solution, but we still argue that it is the best possible one until datasets with better coverage become available.

In order to present the differences of the personalized models a new figure will be added in to the paper and the differences of the models will be discussed by adding a new paragraph in to the manuscript. Furthermore, we will reconstruct a model using the average population data and the differences of between this generic cancer model and personalized models will also be discussed.

(3) On the same note, the authors sought for antimetabolites that are equally affecting all 6 HCC networks. This doesn't prove that it's important to use personalized models as hastily claimed in the results. For a fair comparison, the authors should perform the antimetabolite sensitivity analysis using the data averaged over all 27 HCCs and demonstrate that it leads to a substantially different list of targets. Notably, a diverging result would not confirm that personalized reconstructions are better than averaged ones because we lack any experimental validation to benchmark the prediction. The authors should be more critical on these aspects.

We agree with the comment of the reviewer and criticism about the experimental validation. We will reconstruct a generic HCC model by taking the average of the proteomics data for the 27 HCC patients. We will compare the antimetabolites and discuss the differences of the resulting antimetabolites extensively in the revised version of the manuscript. In order to address the criticism about the experimental validation we will discuss why the specific antimetabolites were not identified in the corresponding personalized model. If we are more clear and critical on these matters we are confident that the reviewer will be satisfied with the results of our study.

(4) The authors claim that the personalized, proteomics-based models "represent a significant improvement over existing...", but there is really no prove for it. In fact, I think that the potential value of such a resource of undersold. Differences are superficially discussed based on GO enrichments, but the peculiarities of the GEMs are hidden under the carpet. In the context of this work, I expect the authors to detail on what pathways are specifically activated or deactivated in each of the 6 HCC of the patients (reconstructed without population data), compared to the reconstruction obtained from averaging all 27 individuals and to hepatocytes. To make a case for personalized network reconstructions, the authors should then emphasize the occurrence of personalized pathway usage, which in turn can be linked to specific sensitivity of individual HCCs to antimetabolites or enzyme inhibition. To sell the GEMs as a resource for the community, more emphasis should be given also to the 83 networks obtained from healthy tissues.

We thank the reviewer for this comment which we indeed agree with. Following this comment we will discuss the activated reactions and pathways in each personalized model as well as in generic HCC model comparing to healthy hepatocytes. This will be included in the revised version of the manuscript. Note though that the work carried out here is based on viewing the metabolic network as a unit rather than as a set of pathways. We try to advocate a more "function centric" view of metabolism; something we do via the concept of metabolic tasks. We agree with that the GO analysis fits badly with this and that a different type of comparison would fit better with the concept of the paper.

(5) I have a few comments on the comparison of antimetabolite inhibition in HCC vs healthy GEMs. (1) Was it based on metabolites that are present in the intersection of all GEMs or their union? The former is correct, the latter causes several trivialities which should be complemented by sound statistics. (2) Among the 83 healthy controls, are there any cell types which are growing/proliferating comparably to a carcinoma?

We used the intersection of the antimetabolites. This will be clarified in the revised version of the manuscript.

We have not looked at the growth of healthy cells specifically, but it is a good suggestion and

something we will look into.

(6)The potential of antimetabolites is overblown. It is misleading to state that "antimetabolites" were identified. The authors predicted only metabolites which are essential in HCCs but not in healthy cells. This is not equivalent to identifying antimetabolites, for instance as Kim et al did with experimental verification. Please tone down and correct the statements throughout the manuscript.

We agree with the reviewer that this could potentially be unclear and we will use "potential antimetabolites" in the entire manuscript. We should note that the concept of "metabolite essentiality" is a related one, but not identical.

(7)It is not clear why antimetabolites should be preferred to specifically inactivating single enzymes. This is a much more intuitive and practicable approach in drug development. Given that the GEMs are available, a computational analysis should be added to compare the two approaches.

Previously, single and double knockouts for inhibiting the growth of the tumors have been tested using genome scale metabolic modeling. Here we focused on the use of antimetabolites, which could potentially be used for inhibition of more than one enzyme enzymes simultaneously. One reason for this is that many of the antimetabolites used clinically in cancer therapy act by blocking many different enzymes (for example several antipurines). A main point in the manuscript is also the in silico testing of toxicity. By studying of similar enzymes might be affected by a certain antimetabolite is could be possible to predict its side effects.

-

Reviewer #2 :

This is a very interesting paper focused on a very important problem, namely how to take advantage of high-throughput data characterizing cancer cell physiology. The authors have extended previous work on an algorithm for developing "personalized" models that integrates data to create functional, predictive computational models. Below are some concerns:

Many grammar issues throughout that must be corrected for clarity of the presentation; e.g., "therefore inferring with the corresponding enzymes" in the Summary...should be "interfering".

We wish to thank Reviewer #2 for detailed reading of our manuscript and providing constructive comments. We will also fix the issues regarding the grammar.

The difference in the # of reactions between the personalized models is surprisingly (to this reviewer) small. Is there any kind of indication about why the variation is at the level it is? What are the primary drivers of the inclusion/exclusion of reactions during the implementation of this tINIT algorithm? Are any of the 67 metabolic "tasks" more important than others for driving the inclusion/exclusion decisions?

Perhaps the authors can speculate on why there is such a small number of reactions unique to any one model. Is this an artifact of the models? Or perhaps there is some biological significance to that observation?

The differences between the models will be presented in a new figure added to the manuscript and the activated reaction differences will be extensively discussed. We think this new section will address the critics of the reviewer about the differences of the personalized models. During the reconstruction of the models we backed up the personalized proteomics data by adding essential biological tasks that occur in every cell type. In order to make sure the uniqueness of the reactions to the model, we will discuss the proteome evidence and will provide more mechanistic explanations for the potential pathways. We should also note that the number of unique enzymes seems to be small as judged by the raw proteomics data (Uhlen, Nat. Biotechnol. 2010; 28: 1248-1250).

The validation of antimetabolites was all done retrospectively. While such a comparison provides some nice support for the model predictions, the paper should be a little more explicit in delineating the workflow (same database from which the protein expression was used to build the models apparently also contained data on which antimetabolites were effective). This doesn't demonstrate

solid statistics for accuracy of the predictions (perhaps the database tested all of the other metabolites that were predicted to be effective but simply did not report).

During the validation of the antimetabolites we checked the Human Metabolome Database and Drugbank and their usage as an antimetabolite in cancer treatment. We also list all clinically used antimetabolites which we could find. We think that the reviewer might have misunderstood something about which databases were used for what. The protein expression was generated in this project as a part of the Human Protein Atlas, and has no association to databases for drugs. This should be clarified in the revised version of the manuscript.

6th paragraph of the antimetabolite section, last sentence, says that "identified as antimetabolites and these metabolites can provide effective treatment strategies"...should probably be "could" or "may" or something else similar. As stated it sounds more like these have been tested. Similar to point in previous paragraph, the authors need to be more explicit about what was predicted, tested, etc. so that claims are not over-stated.

We agree with the reviewer and we will tone down about the usage of terms antimetabolites. We will use it as "potential antimetabolites" in the entire manuscript.

The authors make the frequent claim that their models derived from proteomics data are of a higher quality than models that would have been developed from expression data. However, such a claim, while perhaps attractive, is not necessarily true. Just as one example, the quality of the antibodies for the proteomics data could be variable. Perhaps some simple comparison of the differences or value-added from model generation from the proteomics data could be of value and provide great confidence in the perhaps uniquely high quality of the predictions/ outcomes reported here.

We will discuss the variation of the antibodies in the discussion section and we think the extensive discussion of the model differences will help us to address the reviewer's concern. Note though that expression data is relative while the proteomics data is semi-quantitative. It is therefore not so straight forward to compare. We agree with that it might be too strong to say that these specific models are of higher quality than any others, but we still argue that high-throughput proteomics is the best possible foundation to base GEMs on.

The comparison of the antimetabolite effects on the cancer cells versus their effects on the 83 healthy cell types was not fully developed. What might those results teach us? What were some of the patterns of antimetabolites that were predicted to be effective in cancer but for which there was a potentially high deleterious effect in healthy cell types?

This will be expanded on, as discussed in the answers to the previous reviewer.

-

Reviewer #3:

In their paper "Identification of anticancer drugs for hepatocellular carcinoma through personalized genome-scale metabolic modeling", Agren et al. describe the construction of personalized genome-scale metabolic models (GEM). In the following, the models are used to identify antimetabolites which may be used as anti-cancer drugs.

The paper by Agren et al. describes the next step in the emerging field of human GEMs. In particular the consideration of personalized models is a novel approach which may support the development of individualized treatment designs in the future. The presented work is hence of interest to the MSB readership. There are however, some ambiguities in the MS which need to be addressed by the authors before its final acceptance.

We wish to thank Reviewer #3 for detailed reading of our manuscript and providing constructive comments.

Comments:

i The authors construct 6 GEMs for HCC patients and 83 cell-type specific GEMs. While the latter are only required to perform metabolic functions, the former are additionally tested whether they can additionally form biomass. This bilateral approach presents a fundamental concept in

pharmacology to identify efficient (personalized GEMS) but non-toxic (cell-type specific GEMs) drugs. The authors should discuss their work within the context of efficacy vs. toxicity. Maybe they can even come up with a generic workflow outlining how these two aspects can be distinguished with different kinds of GEMs.

We tested the toxicity of each antimetabolite on healthy cell types of human body using here reconstructed 83 cell type-specific GEMs. This is a completely novel approach and increases the information content by removing the "obvious" hits which would be unlikely to be biologically relevant. It also provides a more detailed understanding of the metabolic effects of a drug/antimetabolite. We agree with the reviewer comments and we will discuss our work in the context of efficacy vs. toxicity

i When constructing the GEMS, the authors probably need some cutoff criteria to distinguish strong, moderate and weak expression. Which cutoff was chosen? Is the structure of the GEMs robust for different value of the cutoff value?

We tested the use of different cutoff criteria during the development of INIT and we will repeat similar analysis for the robustness of the models reconstructed through tINIT algorithm.

i In the last paragraph of the paragraph "Personalized GEMS for HCC patients" the authors discuss the statistics of the GEM construction. Please also include some biochemical/physiological discussion of the pathways which are unique. Is this an artifact of network construction (coverage) or an inherent property of the personalized network model?

We agree with the reviewer and an extensive discussion between the differences between the personalized models and generic HCC model which is reconstructed based on the average population data. The differences on the activated and deactivated reactions as well as their impact on the biological pathway differences will be included in the revised version of the manuscript.

i The same questions arises in the next paragraph "Antimetabolites for HCC patients". Is the finding that some metabolized are only effective in certain patients due to personalized genome structure or due to network coverage during model construction.

In order to avoid the antimetabolite differences due to network coverage, we have used the average population data for the missing protein values in each HCC patients. If we had not included the average population data for the missing points, as suggested by the reviewer#1, then we would have significant differences between the antimetabolites in each HCC patients. In general we agree with the both reviewers that this point should be clarified and extensively discussed in the revised version of the manuscript.

i Figure 1: The fact, that m11 is identified differently than m4 m12 should be visualized differently and not only by a dashed line.

We would like to thank the reviewer for this comment and we will modify the figure.

i The present paper largely covers pharmacodynamic effects of drug therapy. This, however, is only half of the story since pharmacokinetics are equally important for a targeted therapeutic design. GEMs have been integrated before in whole-body PBPK models to address amongst others PK/PD behavior or drug-induced intoxication (which is almost identical to the toxicity test in the manuscript). The authors should discuss their work within the context of this earlier study.

We will include the relevant discussion in to the revised version of the manuscript.

i It is a frequent misconception to consider cellular models (and cellular assays) as an adequate surrogate for real patients. Though the authors avoid this impression in their work, they should nevertheless spend some time discussing the relevance (and limitations) of cellular models for real clinical applications.

We will include the relevant discussion in to the revised version of the manuscript.

i In the same regard, the authors should discuss their GEMS within the context of inter-individual variability. Can variations in the 6 personalized GEMs be compared to inter-individual variability in real patients?

The differences between the models will be presented in a new figure added to the manuscript and the activated reaction differences will be extensively discussed. It is difficult to find comparable statistic, but we agree that we should be able to do more regarding this. We are looking into how the currently used antimetabolites affect different patients.

i Finally, the authors should give some conceptual outlook: What is needed to translate findings in GEMs to the clinic?

The potential usage of the antimetabolites will be discussed.

i Notably, all 6 HCC patients considered are older than 58 years. Is age a bias in the approach? Is more meta-information available regarding the background of the donors? The authors should discuss this.

Unfortunately, we do not have more meta-information for the patients except the age and the sex of the patients. However it should be noted that the 27 patients do not cluster together based on the age and sex of the patients.

2nd Editorial Decision

18 October 2013

Thank you for your message regarding our decision on your manuscript MSB-13-4728. We have now carefully considered the points raised in your appeal letter and point-by-point response to the reviewers' comments.

From our point of view, the most substantial points raised by the reviewers can be summarized as follows:

1. The tINIT approach represents a modification of the existing INIT method rather than a decisive conceptual advance from a methodological point of view.
2. The superiority of the n=1 personalized model compared to an n=27 model is not convincingly demonstrated. In particular, it remains unclear whether personalized models can be conclusively used for patient-level predictions (i.e. differential sensitivity of HCC patient cells to antimetabolites).
3. The activity of the newly identified antimetabolites is not validated experimentally, thus preventing the evaluation of the predictions.

Overall, we acknowledge that using personalized genome-scale metabolic models to account for inter-individual variability when looking for therapeutic targets is potentially a very interesting concept. We also understand the rationale for filling gaps with averaged data. However, we think that point #2 represents an important issue that needs to be more convincingly addressed. In particular, inclusion of additional data demonstrating a correlation between patient-specific data and the predictions from personalized models would considerably strengthen the impact of the study.

While we appreciate that you are willing to address some of the reviewers' comments, at this point we are not convinced that we should revert our previous decision. However, we would be willing to consider editorially a revised and extended version of the manuscript, in which the advantages of personalized models compared to a population-level or generic model are convincingly demonstrated, either by experimentally validating the activity of the newly identified antimetabolites in HCC cells or, maybe even more interestingly, by demonstrating the added value of personalized models for individualized predictions. I am truly sorry to have to disappoint you again but I hope that the comments above can better explain the reasons behind our decision.

On behalf of all the authors, I herewith submit our extended paper entitled "Identification of anticancer drugs for hepatocellular carcinoma through personalized genome-scale metabolic modeling" for publication in *Molecular Systems Biology*. Our presented method allowed for identification of potential therapeutic targets for treatment of HCC by considering individual differences in protein expression patterns and may therefore aid in filling the existing gap between proteomics and drug discovery.

Our paper has been previously reviewed by the referees appointed by *Molecular Systems Biology* and we improved the description of the used methods and the presentation of the personalized proteomics data and genome-scale metabolic models based on their comments. Upon request we will be happy to provide a point-by-point response to the reviewer's comments. We also included new analyses, rewrote a major part of the paper and finally included a new figure, table, supplementary figures and datasets.

In order to develop effective treatment strategies for HCC, we first evaluated the presence/absence of proteins encoded by 15,841 genes in tumors obtained from 27 HCC patients. We identified the differences between six HCC patients based on the proteomics data.

Next, we developed the tINIT (Task-driven Integrative Network Inference for Tissues) algorithm, which allows for automated reconstruction of functional GEMs based on protein evidence and a novel task-driven reconstruction approach.

Thirdly, we applied tINIT to the Human Metabolic Reaction (HMR) 2.0 database together with personalized proteomics data and reconstructed functional personalized GEMs for six HCC patients.

Furthermore, we generated functional GEMs for 83 different healthy cells based on the proteomic data in the Human Protein Atlas (HPA). Finally, we identified 101 antimetabolites that were predicted to inhibit or kill the growth of HCC tumors in all patients. Since, the inhibition of a metabolite may lead to abnormalities in the metabolic functions of a healthy cell, the toxic effect of each antimetabolite was evaluated for a number of metabolic tasks in each of the 83 healthy cell type GEMs. The proposed antimetabolites are therefore likely to damage the tumor in all HCC patients, with the least possible side effects on the function of other healthy cell types. Comprehensive reconstruction of simulation ready personalized GEMs and healthy cell-type GEMs through tINIT algorithm would be a useful resource for the field. GEMs reconstructed through the previously published INIT algorithm could be seen as snapshots of active metabolism in a given cell type, but they were not functional models that could be directly applied for simulations. In this work, we expanded significantly the INIT algorithm in order to allow for direct reconstruction of functional GEMs. The tINIT algorithm allowed us to define a set of core metabolic tasks that should occur in a particular cell type and to reconstruct a functional GEM based on these metabolic tasks. It should be noted that the reconstruction of the functional GEMs is necessary for the application of the methods based on flux balance analysis. Therefore, the tINIT algorithm represent a significant step forward in the reconstruction of cell type-specific models, given that tINIT not only generates connected and consistent metabolic networks, but also ensures functionality by integrating evidence-based metabolic functions which are established for a certain cell type.

Moreover, the tINIT algorithm contains two additional improvements over the original INIT algorithm. Firstly, it constrains the solution so that reversible reactions cannot have flux in both directions simultaneously. This enabled some loops to be included even though they were not connected to the rest of the metabolic network. Secondly, it allows the user the choice of whether net-production of all metabolites should be allowed (which was the case in the original implementation).

We reconstructed personalized GEMs through the use of the tINIT algorithm and we observed notable differences between the reactions and genes during the pair wise comparison of the personalized models. The differences between the numbers of reactions in the personalized models varied between 356 and 610. On the other hand, we observed larger differences in the number of the genes incorporated into the models. The model differences on the present genes varied between 392 and 524, and considering all 2,361 genes shared in all models a 16% to 22% difference was observed between the personalized models. In order to investigate to which extent the personalized models differ from a population model, we also reconstructed a model based on the average HCC

population data. The pair wise reaction and gene differences between the personalized and population models are presented. The model reconstructed based on the averaged model is about as different to the personalized models as they are to each other.

We are also confident that the used method for predicting antimetabolites will be extensively used for future cancer and metabolic diseases related studies. We predicted potential antimetabolites that can inhibit growth of any of the studied HCC tumors and 83% of these identified antimetabolites were effective in disabling growth in all six HCC patients, while 8% were effective in 2-5 patients. 7% of the antimetabolites were only effective in one of the patients. Even though the majority of antimetabolites disable growth of all the HCC tumors, the fact that not all antimetabolites works for all the HCC tumors illustrates the importance of using personalized models rather than relying on generic cancer models. It should be noted that ~15% differences between the personalized models similarly reflected to the ~15% differences between the predicted antimetabolites.

We have also carefully considered the experimental validation of here identified antimetabolites as potential drugs for treatment of HCC at each patient, but found that it is not straight forward to perform experimental validation, except for a full-scale clinical study. In order to evaluate the absence/presence of the proteins between the HCC tumors, we used frozen tissue samples and this limited us for culturing the cells in vitro. We could undertake validation using cell cultures, but considering the targeted approach applied here, we would have to use cell lines closely associated with HCC cancers. In particular, we could use Hep-G2 Hepatocellular carcinoma cell line for validation, but in that case we have to reconstruct a GEM for this specific cell line and this will not represent the personal differences between HCC tumors. Considering the many novel and significant findings from our modeling work, and the very large experimental data set presented in the paper, we have therefore decided to proceed with publication without such an experimental validation. However, we should also note that each antimetabolite was predicted due to the high expression of the associated proteins to the identified metabolites and this was manually checked for each antimetabolite at every patient.

The results of our study can be used to reduce the number of chemical compounds for drug screening by focusing on structural analogs of the identified antimetabolites. Beyond the prediction of new potential drug targets for HCC, the modeling approach presented here may be expanded to study the effect of a standard therapy for a particular individual and hereby evaluate if the treatment is likely to work, and hereby our approach may enable new exciting possibilities for personalized medicine. Considering the above we are confident that our paper is of general interest to researchers in the field of metabolism, systems biology, systems medicine and metabolic network modelling.

3rd Editorial Decision

18 November 2013

Thank you for having submitted a manuscript entitled "Identification of anticancer drugs for HCC through personalized genome-scale metabolic modeling" for consideration for publication in *Molecular Systems Biology*.

First of all, I would like to apologize for the delay in getting back to you, which was due to a recent surge in new manuscript submissions.

Your paper has now been seen by Editors of the Journal, and we have decided to return it to you without sending it for extensive peer review.

In this study, you present a method for predicting potential therapeutic targets for treatment of hepatocellular carcinoma, based on the construction of personalized GEMs using personal proteomics data. We acknowledge that the described approach is interesting and potentially relevant for personalized medicine applications.

Without repeating all the points raised by the referees during the review of the initial version of this manuscript, one of the most fundamental concerns was that the superiority of the presented approach involving personalized models compared to alternative methods (i.e. using an averaged model) is

not convincingly demonstrated. For instance, reviewer #1 had requested "details on what pathways are specifically activated or deactivated in each of the 6 HCC of the patients" and "emphasizing the occurrence of personalized pathway usage". We appreciate that in this revised version you discuss the differences between the individual personalized GEMs in terms of the number of reactions that overlap. But we think that this analysis does not appear to bring further functional insights into the differences between the individual personal models and the potential physiological relevance of these differences.

Similarly, we appreciate that you delineate some of the differences between the antimetabolite predictions obtained using personalized models and the averaged model. The details of the results are not shown (no figure) and the nature, degree and potential explanation of the overlap (or lack thereof) between the antimetabolites predicted by the personalized, the averaged and generic models remain unclear.

Another substantial point raised during the review of this work refers to the fact that the lack of experimental validation of the newly identified antimetabolites precludes a conclusive evaluation of the predictions. We recognize that examining the activity of these antimetabolites in the Hep-G2 hepatocellular carcinoma cell line is not as accurate as performing the validation in patient-derived cell lines. On the other hand, most of the reported antimetabolites were selected because of their predicted general inhibitory activity across all six HCC patients. We agree that Hep-G2 cells are not equivalent to patient-derived HCC cells, but it is also unclear whether this cell line should be dismissed a priori considering the potentially general anti-proliferative effect of the predicted antimetabolites.

Overall, we do acknowledge that using personal omics data to constraint genome-scale metabolic models and derive personalized models is an important concept and we agree that successful use of such models to guide personalized therapies would represent a significant achievement. While we appreciate that in this revised manuscript you have responded to some of the reviewers' comments, we feel that several substantial points remain unaddressed and we think that the discussed advantages and potential applications of the proposed approach for personalized medicine are not yet convincingly demonstrated. As such, I see no other choice than to return the manuscript with the message that we cannot offer to publish it. I am sorry to not be able to bring better news, but I hope that the above comments will explain the reasons behind our decision and will prove to be helpful to you in deciding how to proceed with your work.

Re-submission

14 January 2014

On behalf of all the authors, I herewith submit our extended paper entitled "Identification of anticancer drugs for hepatocellular carcinoma through personalized genome-scale metabolic modeling" for publication at *Molecular Systems Biology*. Our presented method allowed for identification of potential therapeutic targets for treatment of HCC by considering individual differences in protein expression patterns, and may therefore aid in filling the existing gap between proteomics and drug discovery.

Our paper (MSB-13-4728) has been reviewed by three referees appointed by *Molecular Systems Biology* and the referees showed their potential interest to our study. In the recent submission of our revised manuscript (MSB-13-4953), you also acknowledged that using personal omics data to reconstruct genome-scale metabolic models is an important concept and that successful use of such models can be used for guiding personalized therapies.

In this extended version of our manuscript, we improved the description of the used methods and the presentation of the personalized proteomics data and genome-scale metabolic models based on your comments. We included in-vitro experimental validation, new analysis, rewrote major parts of the paper and finally included two new figures and new supplementary datasets. We also addressed each of your specific comments and provided point-by-point responses.

We will be very happy to have your editorial input to make our paper a strong candidate for consideration at *Molecular Systems Biology* and we are looking forward to hearing from you.

Response to the editor's comments:

Thank you for having submitted a manuscript entitled "Identification of anticancer drugs for HCC through personalized genome-scale metabolic modeling" for consideration for publication in Molecular Systems Biology. First of all, I would like to apologize for the delay in getting back to you, which was due to a recent surge in new manuscript submissions. Your paper has now been seen by Editors of the Journal, and we have decided to return it to you without sending it for extensive peer review.

In this study, you present a method for predicting potential therapeutic targets for treatment of hepatocellular carcinoma, based on the construction of personalized GEMs using personal proteomics data. We acknowledge that the described approach is interesting and potentially relevant for personalized medicine applications.

Without repeating all the points raised by the referees during the review of the initial version of this manuscript (MSB-13-4728), one of the most fundamental concerns was that the superiority of the presented approach involving personalized models compared to alternative methods (i.e. using an averaged model) is not convincingly demonstrated. For instance, reviewer #1 had requested "details on what pathways are specifically activated or deactivated in each of the 6 HCC of the patients" and "emphasizing the occurrence of personalized pathway usage". We appreciate that in this revised version you discuss the differences between the individual personalized GEMs in terms of the number of reactions that overlap. But we think that this analysis does not appear to bring further functional insights into the differences between the individual personal models and the potential physiological relevance of these differences.

We have included Dataset 8 into the paper to investigate the activated or deactivated pathways in the personalized and generic HCC models and modified the paper as below.

"The resulting personalized models ranged in size from 4,690 to 4,967 reactions and 1,715 to 2,025 genes. A total of 5,405 reactions and 2,361 genes were shared across the models and 4,212 of the reactions and 1,324 of the genes were present in all six personalized HCC models. It was observed that 248 of the reactions (Figure S3A), 102 of the metabolites (Figure S3B) and 227 of the genes (Figure S3C) were unique to any one model. However, we observed notable differences between the reactions (Figure 3A) and genes (Figure 3B) during a pair wise comparison of the models. The differences between the numbers of reactions in the personalized models varied between 356 and 610 whereas the similarities between the reactions varied between 4,437 and 4,699. On the other hand, we observed larger differences in the number of the genes incorporated into the models. The model differences on the present genes varied between 392 and 524, and considering all 2,361 genes shared in all models a 16% to 22% difference was observed between the personalized models."

"To investigate to which extent the personalized HCC models differ from a HCC population model, we reconstructed a generic HCC model based on the average protein expression of the 27 HCC patients and presented the content of the generic HCC model in Table 1. The pair wise reaction and gene differences between the personalized models and generic HCC model are also presented in Figure 3A and Figure 3B, respectively. As it can be seen, the generic HCC model is about as different to the personalized models as these models are to one another."

"In order to check if a specific pathway is activated in the personalized and generic HCC models, we counted the number of the reactions in the relevant subsystem of HMR2.0 (Dataset 8). The numbers of the reactions in the personalized and generic HCC models did not show any significant differences. We further observed that none of the specific pathways were activated or deactivated in each of the personalized and generic HCC models. However in-depth analysis showed that many reactions in the specific pathways differed between the models, and hence this shows that different enzymes are activated in the different cancers but the same overall metabolic functions are characteristic of HCC."

Similarly, we appreciate that you delineate some of the differences between the antimetabolite predictions obtained using personalized models and the averaged model. The details of the results are not shown (no figure) and the nature, degree and potential explanation of the overlap (or lack thereof) between the antimetabolites predicted by the personalized, the averaged and generic models remain unclear.

We included Figure 4 into the paper for comparing the differences between antimetabolites predicted by the use of personalized and generic HCC models and modified the paper as below.

"By means of our approach, we predicted 147 antimetabolites that can inhibit growth in any of the studied six HCC tumors (Dataset 9). 101 (69%) of these potential antimetabolites were predicted to be effective in disabling growth in all six HCC patients (Figure 4A), 23 (16%) of the antimetabolites were effective in 2-5 patients and the remaining 23 (16%) of the antimetabolites were only effective in one of the patients (Figure 4B). The 46 (31%) of the antimetabolites that are predicted to be

effective in only some of the HCC patients are presented in Figure 4C. Even though 69% of antimetabolites disable growth in all of the HCC tumors, the fact that not all antimetabolites are applicable for all the HCC tumors illustrates the importance of using personalized models rather than relying on a generic HCC model."

"We also predicted the antimetabolites through the use of the generic HCC model that is reconstructed based on the average HCC population data. Analogs of 127 metabolites were predicted as antimetabolites through the generic HCC model and 26 (20%) of these antimetabolites were not suitable for inhibiting the growth in all six HCC patients (Figure 4A). These 26 antimetabolites would therefore not be suitable targets for cancer treatment in all patients. Thus, personalized HCC models allowed us to predict the effect of these false positive antimetabolites on the treatment of all HCC patients, and it is hereby clear that the use of personalized models significantly improve the identification of drug targets effective in all HCC patients."

Another substantial point raised during the review of this work refers to the fact that the lack of experimental validation of the newly identified antimetabolites precludes a conclusive evaluation of the predictions. We recognize that examining the activity of these antimetabolites in the Hep-G2 hepatocellular carcinoma cell line is not as accurate as performing the validation in patient-derived cell lines. On the other hand, most of the reported antimetabolites were selected because of their predicted general inhibitory activity across all six HCC patients. We agree that Hep-G2 cells are not equivalent to patient-derived HCC cells, but it is also unclear whether this cell line should be dismissed a priori considering the potentially general anti-proliferative effect of the predicted antimetabolites.

In our study, antimetabolites were predicted because of their inhibitory activity across all six HCC tumours. Based on your suggestion, we used HepG2 cell line for validation of the predicted antimetabolites as potential anticancer drugs. We included Figure 7 into the paper and included the text below.

"We experimentally evaluated the effect of an L-carnitine analog, one of the predicted antimetabolites for inhibition of HCC tumor growth in all patients. By evaluating proliferation of HepG2 cell lines in presence or absence of Perhexiline maleate salt, we could confirm our genome-scale modeling predictions"

"In order to test the use of an L-carnitine analog as a potential antimetabolite for inhibition of HCC tumor growth, we studied the effect of Perhexiline maleate salt, an inhibitor of carnitine palmitoyltransferase 1 (CPT1) and to a lesser extent CPT2, on the proliferation of the HepG2 cell line. In our study, we used Perhexiline maleate to mimic the effect of the L-carnitine analog since L-carnitine conjugates to FAs and translocates them to the mitochondria through the enzyme CPT1. We treated the HepG2 cells with four different concentrations (2 μ M, 4 μ M, 8 μ M and 20 μ M) of Perhexiline maleate and, determined the viable cells after 24 and 48 hours and compared the inhibitory effect of Perhexiline maleate to Sorafenib (2 μ M and 4 μ M) (Figure 7). Whereas the lower concentrations had no effect, the results clearly showed that the treatment of the HepG2 cell line with 8 μ M and 20 μ M Perhexiline maleate reduced the viability of the HepG2 cell line (Figure 7A), and that its effect is comparable to the effect of Sorafenib on HepG2 cell line (Figure 7B). Lower concentrations of Perhexiline maleate (2 μ M and 4 μ M) appeared not to effect cell viability."

"We reported that the Perhexiline maleate that has been used to mimic the behavior of L-carnitine analog reduces the viability of the HepG2 cell line and thereby validated the use of genome-scale modeling based predictions on anticancer drug identification."

Overall, we do acknowledge that using personal omics data to constraint genome-scale metabolic models and derive personalized models is an important concept and we agree that successful use of such models to guide personalized therapies would represent a significant achievement. While we appreciate that in this revised manuscript you have responded to some of the reviewers' comments, we feel that several substantial points remain unaddressed and we think that the discussed advantages and potential applications of the proposed approach for personalized medicine are not yet convincingly demonstrated. As such, I see no other choice than to return the manuscript with the message that we cannot offer to publish it. I am sorry to not be able to bring better news, but I hope that the above comments will explain the reasons behind our decision and will prove to be helpful to you in deciding how to proceed with your work.

"In conclusion, we identified potential antimetabolites which may inhibit the growth of HCC tumors through the use of personalized metabolic modeling, proposed the usage of antimetabolites for the treatment of HCC and demonstrated the inhibitory effect of the L-carnitine analog, one of the predicted antimetabolites, on the proliferation of the HepG2 cell line. The results of our study can be used to reduce the number of chemical compounds for drug screening by focusing on the structural analogs of the identified antimetabolites. Beyond the prediction of new potential drug targets for

HCC, the modeling approach presented here may be expanded to study the effect of a standard therapy for a particular individual and hereby evaluate if the treatment is likely to work, and hereby our approach may enable new exciting possibilities for personalized medicine." Considering the above we are confident that our paper is of general interest to researchers in the field of metabolism, systems biology, systems medicine and metabolic network modelling.

Response to the reviewers' comments

Reviewer #1:

Agren et al use a detailed data set of protein expression (reduced to the binary case) of 27 hepatocellular carcinoma isolated from patients. For 6 samples, they generated a minimal metabolic network by pruning the complete HMR 2.0 model subject to the boundaries imposed by the expression data and a set of predefined metabolic capabilities that the minimal network must fulfill. The approach builds on a previous method by the same group and is named tINIT.

In order to identify compounds to efficiently and selectively treat HCC, the query the metabolic reconstructions for metabolites whose removal impairs fulfillment of the predefined metabolic capabilities in tumors but not in reconstruction of healthy cell types. For this they postulate that for every metabolite present in the network, it is possible to generate a variant (antimetabolite) which is metabolically inert and capable of virtually silencing all enzymes that use the metabolite as substrate. About hundred metabolites in a dozen pathways could be proposed. The predictions are discussed at large with literature data, but no experimental validation presented.

We wish to thank Reviewer #1 for detailed reading of our manuscript and providing constructive comments. A small point is that the expression scores aren't reduced to binary and that the method is not about finding the minimal set of reactions which include the positive scoring reactions (this would be similar to the GIMME algorithm). Rather, the objective is to find a network which maximizes the sum of scores while being capable of performing all the metabolic tasks. This is a more flexible and complex task.

In our study, antimetabolites were predicted because of their inhibitory activity across all six HCC tumours. Based on the suggestion of the editors, we used HepG2 cell line for validation of the predicted antimetabolites as potential anticancer drugs. We included Figure 7 into the paper and included the text below in the relevant section.

"We experimentally evaluated the effect of an L-carnitine analog, one of the predicted antimetabolites for inhibition of HCC tumor growth in all patients. By evaluating proliferation of HepG2 cell lines in presence or absence of Perhexiline maleate salt, we could confirm our genome-scale modeling predictions"

"In order to test the use of an L-carnitine analog as a potential antimetabolite for inhibition of HCC tumor growth, we studied the effect of Perhexiline maleate salt, an inhibitor of carnitine palmitoyltransferase 1 (CPT1) and to a lesser extent CPT2, on the proliferation of the HepG2 cell line. In our study, we used Perhexiline maleate to mimic the effect of the L-carnitine analog since L-carnitine conjugates to FAs and translocates them to the mitochondria through the enzyme CPT1. We treated the HepG2 cells with four different concentrations (2 μ M, 4 μ M, 8 μ M and 20 μ M) of Perhexiline maleate and, determined the viable cells after 24 and 48 hours and compared the inhibitory effect of Perhexiline maleate to Sorafenib (2 μ M and 4 μ M) (Figure 7). Whereas the lower concentrations had no effect, the results clearly showed that the treatment of the HepG2 cell line with 8 μ M and 20 μ M Perhexiline maleate reduced the viability of the HepG2 cell line (Figure 7A), and that its effect is comparable to the effect of Sorafenib on HepG2 cell line (Figure 7B). Lower concentrations of Perhexiline maleate (2 μ M and 4 μ M) appeared not to effect cell viability." "We reported that the Perhexiline maleate that has been used to mimic the behavior of L-carnitine analog reduces the viability of the HepG2 cell line and thereby validated the use of genome-scale modeling based predictions on anticancer drug identification."

Major comments:

(1) A pillar of the manuscript is the development of the tINIT method for the reconstruction. Albeit it constitutes a minor improvement of a previous method, I like the idea. However, the manuscript fails

to test its merits. I had a cursory glance at the tasks in the supplement to understand. My impression is that the list is arbitrary and also redundant. A few examples: Why are cells supposed to consume non-essential amino acids? Why are cells assumed to synthesize intermediary compounds (e.g. glycolytic compounds) while these are anyway necessary for the task of growing? Why are cells obliged to grow on HAM's medium while they were sampled from biopsies? What is the sense of constraining mitochondrial acetyl-CoA de novo synthesis by the "acetyl-CoA[m] => CoA[m]" reaction?

To appreciate the novelty of the method and its benefits over previous, the authors should (1) explain the rationale of the different tasks; (2) compare the results obtained with and without tasks (tINIT vs INIT); (3) systematically compare the impact and independency of single tasks.

The rationale behind the metabolic tasks was to include all functions for which we had literature evidence that they can be performed by all cell types. It was a major oversight that the reference list wasn't present in the submitted Supplementary material, but this has been fixed now. The reason for having partly redundant tasks is twofold: 1) We wanted to have a comprehensive list of functions which all cells can perform in order to have a more fine grained view of how different antimetabolites affect the metabolic capabilities. An example would be in the point raised by the reviewer above, where all cells can synthesize glycolytic compounds but only the cancer cells must be able to use them for growth. 2) It is a computational advantage to split more complex task (such as growth) into its simpler constituents (such as biosynthesis of precursors) due to how tINIT is implemented. It also gives a greater understanding of which reactions are added for which tasks (rather than just add a bunch of reactions to enable growth in one step). Redundant tasks are no issue since it's the union of the essential reactions that are "forcefully" included in the network. Adding the same, or partly overlapping, tasks several times will have no effect on the output.

We first removed the consumption of non-essential amino acids from the list of the metabolic tasks and included the text below.

"The tasks used for imposing constraints on the functionality of the reconstructed models (see Dataset 7) are based on metabolic functions which are known to occur in all cell types. As such there is some redundancy between the tasks. For example, ADP re-phosphorylation is a prerequisite for the biosynthesis of some of the precursors (which in turn is a prerequisite for biomass formation). The reason for this is twofold. On the one hand it makes for a less computationally demanding optimization, as the reconstruction can be performed in a more stepwise manner. On the other hand it makes for more fine-grained analysis of the impact of each task, in particular when it comes to the effect of antimetabolites. It should be noted that the redundancy is not a problem from a reconstruction viewpoint, since the resulting GEM will look the same regardless."

The reason for using HAM's medium is that it is a well-defined medium that enables growth of many cell types. It has also been used for the same purpose by other groups when simulating cancer growth in vivo using GEMs. If we had used, say, metabolomics of blood or tissue instead the same question could be asked there.

De novo synthesis of mitochondrial acetyl-CoA is expressed like that because "de novo" refers here to the activated acetyl group rather than to the acetyl-CoA. The difference is that with a formulation like "acetyl-CoA[m] => acetyl-CoA[excreted]" the network would also need to have CoA synthesis. This is a separate task, and since it's a co-factor it isn't synthesized in any large amounts.

We reconstructed the personalized models with INIT and tINIT algorithms and personalized models generated by INIT algorithm were not able to grow. We have not included the statistics for models generated by INIT at this stage and we clarified the differences between the two algorithms by adding the text below.

"Previously, Agren et al. (2012) developed the INIT (Integrative Network Inference for Tissues) algorithm for automated generation of cell type-specific and cancer GEMs. These networks could be seen as snapshots of active metabolism in a given cell type, but they were not functional models that could be directly applied for simulations. In this work, we expanded significantly the INIT algorithm in order to allow for direct reconstruction of functional GEMs.

The tINIT algorithm allows the user to define metabolic tasks which the resulting model should be able to perform. These metabolic tasks can be outlined and used as an input to tINIT algorithm in Microsoft Excel for convenience (Dataset 7). The algorithm then works by first identifying the set of reactions in the generic model which, if any of them are excluded, cause one or more of the tasks to fail. This set of reactions then have to be in the resulting model. Note that this is not the same thing as the smallest set required for performing the tasks, as there can be iso-enzymes or alternative pathways. The INIT algorithm is then implemented as described in the original paper (Agren et al, 2012), but with the additional constraint that these reactions have to be in the solution. The resulting

solution has to adjust to fit with these reactions, and is therefore likely to be close to being able to perform the tasks. In a final step each task is tested in a sequential manner, and if it cannot be performed then the gap-filling algorithm in the RAVEN Toolbox (Agren et al, 2013) is applied in order to enable it. The sequential testing means that the order of the tasks could theoretically impact which reactions are included in the gap-filling step. However, this is normally not the case, since the solution is close to functional because of the set of required reactions.

tINIT contains two additional improvements over the original INIT algorithm. Firstly, it constrains the solution so that reversible reactions cannot have flux in both directions simultaneously. This enabled some loops to be included even though they were not connected to the rest of the metabolic network. Secondly, it allows the user the choice of whether net-production of all metabolites should be allowed (which was the case in the original implementation). The tINIT algorithm is implemented and extensively commented in the RAVEN Toolbox (Agren et al, 2013) together with functions for working with the concept of metabolic tasks (www.sysbio.se/BioMet)."

(2)The second cornerstone of the manuscript is the personalized genome-scale modeling. Important claims are made, but there are several flaws. First of all, the reconstruction is based on population-averaged data. It is not clear to me why so many missing values are present or what the cause for this is. Nevertheless, using median data from the entire population to fill gaps is a killer. The authors should refrain from it, and use only the proteomic data obtained for the 6 patients with dense measurements. This is a precondition to claim that the reconstruction was made based on n=1. This is likely going to reduce the size of the reconstructed models, but better reflects the data that one would get from a single patient.

We evaluated the presence/absence of proteins encoded by 15,841 genes in HCC tumors obtained from 27 patients using immunohistochemistry and presented personalized proteomics data. However, we reconstructed models only for the six HCC patients with the best coverage (between 9,312 and 14,561 genes measured). Also note that only a subset of these genes code for enzymes. This still constitutes a very large coverage compared to other published data sets.

We disagree with this comment of reviewer #1 about the usage of the average HCC data for the missing protein evidence. We discussed this internally during the preparation of manuscript and we think it is more accurate to use average data rather than using no evidence. This is a standard approach in many bioinformatics applications, and could be viewed as making the most of the available data. The fact that some small proportion of the measurements was based on population does not render the models "non-personalized". Also, the way that tINIT is designed; it would not be a good solution to simply reconstruct models without the population data, as it would rely on some arbitrarily chosen value instead.

Another important reason is that we use model differences during the identification of the antimetabolites, and it would be a bigger bias to compare models for which different sets of proteins had been measured. We agree with the reviewer in that this is not an ideal solution, but we still argue that it is the best possible one until datasets with better coverage become available. We included the text below into the manuscript.

"Since the set of measured proteins differed somewhat between the six HCC patients, averaged data from all 27 HCC patients was used to fill the gaps. tINIT requires that all reactions in the reference network are given a score, and the alternative solution would be to use an arbitrary negative score for proteins which were not measured in some given patient. This would represent a larger bias, and it was therefore chosen that averaged data should be used for the missing protein expression value. It should be also noted that the filled protein expression data is relatively small comparing to the measured protein expressions in each patient."

In order to present the differences of the personalized models a new figure was added in to the paper and the text below was included.

"The resulting personalized models ranged in size from 4,690 to 4,967 reactions and 1,715 to 2,025 genes. A total of 5,405 reactions and 2,361 genes were shared across the models and 4,212 of the reactions and 1,324 of the genes were present in all six personalized HCC models. It was observed that 248 of the reactions (Figure S3A), 102 of the metabolites (Figure S3B) and 227 of the genes (Figure S3C) were unique to any one model. However, we observed notable differences between the reactions (Figure 3A) and genes (Figure 3B) during a pair wise comparison of the models. The differences between the numbers of reactions in the personalized models varied between 356 and 610 whereas the similarities between the reactions varied between 4,437 and 4,699. On the other hand, we observed larger differences in the number of the genes incorporated into the models. The

model differences on the present genes varied between 392 and 524, and considering all 2,361 genes shared in all models a 16% to 22% difference was observed between the personalized models."

"To investigate to which extent the personalized HCC models differ from a HCC population model, we reconstructed a generic HCC model based on the average protein expression of the 27 HCC patients and presented the content of the generic HCC model in Table 1. The pair wise reaction and gene differences between the personalized models and generic HCC model are also presented in Figure 3A and Figure 3B, respectively. As it can be seen, the generic HCC model is about as different to the personalized models as these models are to one another."

(3)On the same note, the authors sought for antimetabolites that are equally affecting all 6 HCC networks. This doesn't prove that it's important to use personalized models as hastily claimed in the results. For a fair comparison, the authors should perform the antimetabolite sensitivity analysis using the data averaged over all 27 HCCs and demonstrate that it leads to a substantially different list of targets. Notably, a diverging result would not confirm that personalized reconstructions are better than averaged ones because we lack any experimental validation to benchmark the prediction. The authors should be more critical on these aspects.

We included Figure 4 into the paper for comparing the differences between antimetabolites predicted by the use of personalized and generic HCC models and modified the paper as below.

"By means of our approach, we predicted 147 antimetabolites that can inhibit growth in any of the studied six HCC tumors (Dataset 9). 101 (69%) of these potential antimetabolites were predicted to be effective in disabling growth in all six HCC patients (Figure 4A), 23 (16%) of the antimetabolites were effective in 2-5 patients and the remaining 23 (16%) of the antimetabolites were only effective in one of the patients (Figure 4B). The 46 (31%) of the antimetabolites that are predicted to be effective in only some of the HCC patients are presented in Figure 4C. Even though 69% of antimetabolites disable growth in all of the HCC tumors, the fact that not all antimetabolites are applicable for all the HCC tumors illustrates the importance of using personalized models rather than relying on a generic HCC model."

"We also predicted the antimetabolites through the use of the generic HCC model that is reconstructed based on the average HCC population data. Analogs of 127 metabolites were predicted as antimetabolites through the generic HCC model and 26 (20%) of these antimetabolites were not suitable for inhibiting the growth in all six HCC patients (Figure 4A). These 26 antimetabolites would therefore not be suitable targets for cancer treatment in all patients. Thus, personalized HCC models allowed us to predict the effect of these false positive antimetabolites on the treatment of all HCC patients, and it is hereby clear that the use of personalized models significantly improve the identification of drug targets effective in all HCC patients."

(4)The authors claim that the personalized, proteomics-based models "represent a significant improvement over existing...", but there is really no prove for it. In fact, I think that the potential value of such a resource of undersold. Differences are superficially discussed based on GO enrichments, but the peculiarities of the GEMs are hidden under the carpet. In the context of this work, I expect the authors to detail on what pathways are specifically activated or deactivated in each of the 6 HCC of the patients (reconstructed without population data), compared to the reconstruction obtained from averaging all 27 individuals and to hepatocytes. To make a case for personalized network reconstructions, the authors should then emphasize the occurrence of personalized pathway usage, which in turn can be linked to specific sensitivity of individual HCCs to antimetabolites or enzyme inhibition.

To sell the GEMs as a resource for the community, more emphasis should be given also to the 83 networks obtained from healthy tissues.

We thank the reviewer for this comment which we indeed agree with. Following this comment, we presented the differences of the personalized models and generic HCC model. We also included Dataset 8 into the paper to investigate the activated or deactivated pathways in the personalized and generic HCC models and modified the paper as below.

"In order to check if a specific pathway is activated in the personalized and generic HCC models, we counted the number of the reactions in the relevant subsystem of HMR2.0 (Dataset 8). The numbers of the reactions in the personalized and generic HCC models did not show any significant differences. We further observed that none of the specific pathways were activated or deactivated in each of the personalized and generic HCC models. However in-depth analysis showed that many

reactions in the specific pathways differed between the models, and hence this shows that different enzymes are activated in the different cancers but the same overall metabolic functions are characteristic of HCC."

(5) I have a few comments on the comparison of antimetabolite inhibition in HCC vs healthy GEMs. (1) Was it based on metabolites that are present in the intersection of all GEMs or their union? The former is correct; the latter causes several trivialities which should be complemented by sound statistics. (2) Among the 83 healthy controls, are there any cell types which are growing / proliferating comparably to a carcinoma?

We used the intersection of the antimetabolites. This is clarified in the revised version of the manuscript. None of the healthy cells can grow under the same conditions as for the cancer cells.

(6) The potential of antimetabolites is overblown. It is misleading to state that "antimetabolites" were identified. The authors predicted only metabolites which are essential in HCCs but not in healthy cells. This is not equivalent to identifying antimetabolites, for instance as Kim et al did with experimental verification. Please tone down and correct the statements throughout the manuscript.

We agree with the reviewer that this could potentially be unclear and we used "potential antimetabolites" in the entire manuscript. We should note that the concept of "metabolite essentiality" is a related one, but not identical.

(7) It is not clear why antimetabolites should be preferred to specifically inactivating single enzymes. This is a much more intuitive and practicable approach in drug development. Given that the GEMs are available, a computational analysis should be added to compare the two approaches.

Previously, single and double knockouts for inhibiting the growth of the tumors have been tested using genome-scale metabolic modeling. Here we focused on the use of antimetabolites, which could potentially be used for inhibition of more than one enzyme simultaneously. One reason for this is that many of the antimetabolites used clinically in cancer therapy act by blocking many different enzymes (for example several antipurines). A main point in the manuscript is also the in silico testing of toxicity and potential side effects, which is made possible by focusing on antimetabolites.

Reviewer #2 :

This is a very interesting paper focused on a very important problem, namely how to take advantage of high-throughput data characterizing cancer cell physiology. The authors have extended previous work on an algorithm for developing "personalized" models that integrates data to create functional, predictive computational models. Below are some concerns:

Many grammar issues throughout that must be corrected for clarity of the presentation; e.g., "therefore inferring with the corresponding enzymes" in the Summary...should be "interfering".

We wish to thank Reviewer #2 for detailed reading of our manuscript and providing constructive comments. We fixed issues regarding the grammar in the revised manuscript. Some of the reviewer's comments were also expanded on, as discussed in the answers to the Reviewer #1. Please also see our response to the Reviewer #1.

The difference in the # of reactions between the personalized models is surprisingly (to this reviewer) small. Is there any kind of indication about why the variation is at the level it is? What are the primary drivers of the inclusion/exclusion of reactions during the implementation of this tINIT algorithm? Are any of the 67 metabolic "tasks" more important than others for driving the inclusion/exclusion decisions?

In order to present the differences of the personalized models a new figure was added in to the paper and the text below was included.

"The differences between the numbers of reactions in the personalized models varied between 356 and 610 whereas the similarities between the reactions varied between 4,437 and 4,699. On the other hand, we observed larger differences in the number of the genes incorporated into the models. The

model differences on the present genes varied between 392 and 524, and considering all 2,361 genes shared in all models a 16% to 22% difference was observed between the personalized models."

This was expanded on, as discussed in the answers to the previous reviewer.

Perhaps the authors can speculate on why there is such a small number of reactions unique to any one model. Is this an artifact of the models? Or perhaps there is some biological significance to that observation?

The differences between the models were presented in a new figure added to the manuscript. We think this new section addressed the critics of the reviewer about the differences of the personalized models. During the reconstruction of the models we backed up the personalized proteomics data by adding essential biological tasks that occur in every cell type. Regarding the uniqueness of the reactions to the model, we should also note that the number of unique enzymes seems to be small as judged by the raw proteomics data (Uhlen, Nat. Biotechnol. 2010; **28**: 1248–1250) as well.

The validation of antimetabolites was all done retrospectively. While such a comparison provides some nice support for the model predictions, the paper should be a little more explicit in delineating the workflow (same database from which the protein expression was used to build the models apparently also contained data on which antimetabolites were effective). This doesn't demonstrate solid statistics for accuracy of the predictions (perhaps the database tested all of the other metabolites that were predicted to be effective but simply did not report).

During the validation of the antimetabolites we checked the Human Metabolome Database and Drugbank for their usage as antimetabolites in cancer treatment. We also list all clinically used antimetabolites that we could find. We agree that it would be valuable to also have information on "failed" targets, but such data is difficult to come by.

The protein expression was generated in this project as a part of the Human Protein Atlas, and has no association to databases for drugs. This was clarified in the revised version of the manuscript.

6th paragraph of the antimetabolite section, last sentence, says that "identified as antimetabolites and these metabolites can provide effective treatment strategies"...should probably be "could" or "may" or something else similar. As stated it sounds more like these have been tested. Similar to point in previous paragraph, the authors need to be more explicit about what was predicted, tested, etc. so that claims are not over-stated.

We agree with the reviewer and we tone

d down about the usage of terms antimetabolites. We used as "potential antimetabolites" or "predicted antimetabolites" in the entire manuscript.

The authors make the frequent claim that their models derived from proteomics data are of a higher quality than models that would have been developed from expression data. However, such a claim, while perhaps attractive, is not necessarily true. Just as one example, the quality of the antibodies for the proteomics data could be variable. Perhaps some simple comparison of the differences or value-added from model generation from the proteomics data could be of value and provide great confidence in the perhaps uniquely high quality of the predictions/ outcomes reported here.

We agree with the reviewer's comments and the variation of the antibodies used for generation of the proteomics data have been extensively discussed in the main Human Protein Atlas paper (Uhlen, Nat. Biotechnol. 2010; **28**: 1248–1250). We think the extensive discussion of the model differences helped us to address the reviewer's concern. Note though that expression data is relative while the proteomics data is semi-quantitative. It is therefore not so straight forward to compare. We agree with that it might be too strong to say that these specific models are of higher quality than any others, but we still argue that high-throughput proteomics is the best possible foundation to base GEM reconstruction on.

The comparison of the antimetabolite effects on the cancer cells versus their effects on the 83 healthy cell types was not fully developed. What might those results teach us? What were some of the patterns of antimetabolites that were predicted to be effective in cancer but for which there was a potentially high deleterious effect in healthy cell types?

This was expanded on, as discussed in the answers to the previous reviewer.

Reviewer #3 :

In their paper "Identification of anticancer drugs for hepatocellular carcinoma through personalized genome-scale metabolic modeling", Agren et al. describe the construction of personalized genome-scale metabolic models (GEM). In the following, the models are used to identify antimetabolites, which may be used as anti-cancer drugs.

The paper by Agren et al. describes the next step in the emerging field of human GEMs. In particular the consideration of personalized models is a novel approach which may support the development of individualized treatment designs in the future. The presented work is hence of interest to the MSB readership. There are however, some ambiguities in the MS which need to be addressed by the authors before its final acceptance.

We wish to thank Reviewer #3 for detailed reading of our manuscript and providing constructive comments. Some of the reviewer's comments were also expanded on, as discussed in the answers to the Reviewer #1. Please also see our response to Reviewer #1.

Comments:

- The authors construct 6 GEMs for HCC patients and 83 cell-type specific GEMs. While the latter are only required to perform metabolic functions, the former are additionally tested whether they can additionally form biomass. This bilateral approach presents a fundamental concept in pharmacology to identify efficient (personalized GEMs) but non-toxic (cell-type specific GEMs) drugs. The authors should discuss their work within the context of efficacy vs. toxicity. Maybe they can even come up with a generic workflow outlining how these two aspects can be distinguished with different kinds of GEMs.*

We tested the toxicity of each antimetabolite on healthy cell types of human body using here reconstructed 83 cell type-specific GEMs. This is a novel approach and increases the information content by removing the "obvious" hits which would be unlikely to be biologically relevant. It also provides a more detailed understanding of the metabolic effects of a drug/antimetabolite.

"In this study, we identified potential antimetabolites that were predicted to inhibit or kill the growth of HCC tumors in all six patients (Figure 1C). Since the inhibition of a metabolite may lead to abnormalities in the metabolic functions of a healthy cell, the toxic effect of each antimetabolite was evaluated for a number of metabolic tasks in each of the 83 healthy cell type GEMs. The proposed antimetabolites are therefore likely to damage the tumor in all HCC patients, with the least possible side effects on the function of other healthy cell types."

"The analogs of L-carnitine and metabolites involved in the synthesis of L-carnitine were proposed as antimetabolites for the treatment of all six HCC patients due to the non-toxic effect to here studied healthy cell types (Figure 6)."

- When constructing the GEMs, the authors probably need some cutoff criteria to distinguish strong, moderate and weak expression. Which cutoff was chosen? Is the structure of the GEMs robust for different value of the cutoff value?*

We evaluated the use of different cut-off criteria during the development of INIT (Agren et al, 2012). We repeated similar analysis for the robustness of the models reconstructed through tINIT algorithm and get very similar GEMs. Also note that the incorporation of metabolic tasks in the reconstruction results in a smaller solution space and more constrained models. The robustness with respect to cut-off values might seem somewhat counterintuitive given the very high dimensionality of GEMs. One possible interpretation for this is that the network can be viewed as consisting of a set of interconnected "modules" (such as a pathway, a metabolic task or some other functionality). Each module can then have only a few degrees of freedom. Consider for example the synthesis of isopentenyl pyrophosphate (in bacteria, for illustration purposes). That could be done via the mevalonate pathway or via the MEP/DOXP pathway. Since both these pathways involve intermediates which are unique or sparsely used elsewhere (HMG-CoA and DOXP for example), the pathways have to be included/excluded *as a whole*. This corresponds to that the algorithm only has to choose between four options; include one pathway, include the other, include both or exclude

both. These numerous short linear dependencies between reactions serve to average out the gene scores, thereby reducing the importance of the chosen cut-off values.

• In the last paragraph of the paragraph "Personalized GEMS for HCC patients" the authors discuss the statistics of the GEM construction. Please also include some biochemical/physiological discussion of the pathways which are unique. Is this an artifact of network construction (coverage) or an inherent property of the personalized network model?

This was expanded on, as discussed in the answers to the previous reviewers. Please also see our response to Reviewer #1.

• The same questions arises in the next paragraph "Antimetabolites for HCC patients". Is the finding that some metabolized are only effective in certain patients due to personalized genome structure or due to network coverage during model construction?

In order to avoid the antimetabolite differences due to network coverage, we have used the average population data for the missing protein values in each HCC patients. If we had not included the average population data for the missing points, as suggested by the reviewer#1, then we would have significant differences between the antimetabolites in each HCC patients. This point was clarified and extensively discussed in the revised version of the manuscript. Please also see our response to Reviewer #1.

• Figure 1: The fact, that m11 is identified differently than m4 m12 should be visualized differently and not only by a dashed line.

We would like to thank the reviewer for this comment. We modified the figure and included a very detailed figure legend.

• The present paper largely covers pharmacodynamic effects of drug therapy. This, however, is only half of the story since pharmacokinetics are equally important for a targeted therapeutic design. GEMs have been integrated before in whole-body PBPK models to address amongst others PK/PD behavior or drug-induced intoxication (which is almost identical to the toxicity test in the manuscript). The authors should discuss their work within the context of this earlier study. It is a frequent misconception to consider cellular models (and cellular assays) as an adequate surrogate for real patients. Though the authors avoid this impression in their work, they should nevertheless spend some time discussing the relevance (and limitations) of cellular models for real clinical applications.

We find the comment of the reviewer very interesting. However, this paper doesn't deal with whole-body modelling, but rather focuses on in silico predictions of cellular responses to perturbations. As such we feel that this discussion is outside of the scope of the paper, particularly as it is already rather broad. We have recently discussed the use of PBPK models and GEMs for modeling of whole body metabolism (Mardinoglu et al., Biotechnol. Journal, 2013, 9, 985).

We think GEMs are very useful resources for the reconstruction of PBPK models and this has been demonstrated by a recent paper by Krauss et al. (PLOSComput. Biol. 2012, 8, e1002750).

We have predicted antimetabolites using genome-scale metabolic modeling and tested the effect of L-carnitine in the growth of HepG2 cell line. Further in vivo testing is required for validating the prediction, as well as for testing the pharmacokinetic effects.

• In the same regard, the authors should discuss their GEMS within the context of inter-individual variability. Can variations in the 6 personalized GEMs be compared to inter-individual variability in real patients?

The differences between the models were presented in a new figure added to the manuscript. Please also see our response to Reviewer #1. We should also note that we obtained comparable proteomics differences and the models differences for each patient.

• Finally, the authors should give some conceptual outlook: What is needed to translate findings in GEMs to the clinic?

"We reported that the Perhexiline maleate that has been used to mimic the behavior of L-carnitine analog reduces the viability of the HepG2 cell line and thereby validated the use of genome-scale modeling based predictions on anticancer drug identification."

"The results of our study can be used to reduce the number of chemical compounds for drug screening by focusing on the structural analogs of the identified antimetabolites. Beyond the prediction of new potential drug targets for HCC, the modeling approach presented here may be expanded to study the effect of a standard therapy for a particular individual and hereby evaluate if the treatment is likely to work, and hereby our approach may enable new exciting possibilities for personalized medicine."

• *Notably, all 6 HCC patients considered are older than 58 years. Is age a bias in the approach? Is more meta-information available regarding the background of the donors? The authors should discuss this.*

Unfortunately, we do not have more meta-information for the patients except the age and the sex of the patients. However it should be noted that the 27 patients do not cluster together based on the age and sex of the patients and we do therefore not think that there is any bias. We are therefore confident that we could reconstruct personalized models rather than population based models. It will, however, as the reviewer point be very interesting in the future to combine GEMs with more extensive metadata, such as blood chemistry, but this may require dedicated clinical studies and hence outside the scope of the present paper.

4th Editorial Decision

17 February 2014

Thank you again for submitting your work to Molecular Systems Biology. We have now heard back from the two referees who agreed to evaluate your manuscript. As you will see from the reports below, the referees think that their previous concerns have been satisfactorily addressed and are now positive on your study. Reviewer #1 lists a few points, which we would ask you to address in a revision of the manuscript.

Please resubmit your revised manuscript online, with a covering letter listing amendments and responses to each point raised by the referees. Please resubmit the paper ****within one month**** and ideally as soon as possible. If we do not receive the revised manuscript within this time period, the file might be closed and any subsequent resubmission would be treated as a new manuscript. Please use the Manuscript Number (above) in all correspondence.

REFEREE REPORTS

Reviewer #1:

The authors have adequately addressed most of the points raised in the last review.

Comments:

- "Patient" is used throughout the manuscript even though the work deals with cellular networks (see previous review). The limitations of using cellular models for the design of therapeutic treatment strategies in human beings should be discussed at any rate.
- Figure 1A: In addition to "antimetabolites", "efficacy" should be used for the labelling the left column. Efficacy (i.e. inhibition of tumor growth) should also be introduced in the main text.

Reviewer #2:

I believe the authors have adequately addressed the previous critiques. The quality of the experimental validation of the specific antimetabolite is not extensive, but I believe the overall contribution is worthwhile.

We wish to thank both reviewers for detailed reading of our manuscript.

Reviewer #1:

The authors have adequately addressed most of the points raised in the last review.

Comments:

- *"Patient" is used throughout the manuscript even though the work deals with cellular networks (see previous review). The limitations of using cellular models for the design of therapeutic treatment strategies in human beings should be discussed at any rate.*

The limitations of using cellular models for the design of therapeutic treatment strategies in human beings is discussed and marked as red on page 11.

*"It is of course important to note that the findings presented here are based on cellular models, and do not take systemic effects into consideration. One way to alleviate this could be to integrate the method described here with whole-body pharmacokinetic and pharmacodynamics modeling."
- Figure 1A: In addition to "antimetabolites", "efficacy" should be used for the labelling the left column. Efficacy (i.e. inhibition of tumor growth) should also be introduced in the main text.*

We have modified the Figure 1A and introduced efficacy in the main text. Marked as red on page 4. "(i.e., having high efficacy and low toxicity)"

Reviewer #2:

I believe the authors have adequately addressed the previous critiques. The quality of the experimental validation of the specific antimetabolite is not extensive, but I believe the overall contribution is worthwhile.